# TERRACODEC: COMPRESSING OPTICAL EARTH OBSERVATION DATA

## ABSTRACT

Earth observation (EO) satellites produce massive streams of multispectral image time series, posing pressing challenges for storage and transmission. Yet, learned EO compression remains fragmented and lacks publicly available, large-scale pretrained codecs. Moreover, prior work has largely focused on image compression, leaving temporal redundancy and EO video codecs underexplored. To address these gaps, we introduce *TerraCodec* (TEC), a family of learned codecs pretrained on Sentinel-2 EO data. TEC includes efficient multispectral image variants and a Temporal Transformer model (TEC-TT) that leverages dependencies across time. To overcome the fixed-rate setting of today's neural codecs, we present Latent Repacking, a novel method for training flexible-rate transformer models that operate on varying rate-distortion settings. TerraCodec outperforms classical codecs, achieving $3 - 10 \times$ stronger compression at equivalent image quality. Beyond compression, TEC-TT enables zero-shot cloud inpainting, surpassing state-of-the-art methods on the AllClear benchmark. Our results establish EO-trained neural codecs and temporal compression as a promising direction for Earth observation. Code and model weights will be released under a permissive license.

## 1 INTRODUCTION

The exponential growth of Earth Observation (EO) data, driven by initiatives such as the Copernicus program, creates critical bottlenecks in storage, transmission, and processing (Guo et al., 2016; Wilkinson et al., 2024). EO imagery also differs fundamentally from natural images. It is multispectral, with up to dozens of channels beyond the visible RGB spectrum; and multi-temporal, with images captured at regular intervals from near-constant viewpoints but subject to seasonal, atmospheric, and cloud cover changes. As a result, EO scenes contain strong spatial and spectral redundancy, while temporal evolution arises from recurring seasonal patterns rather than object motion. These properties create compression challenges distinct from natural images, but make EO well-suited for learned approaches that capture domain-specific priors (Gomes et al., 2025). Despite advances in neural codecs for natural images and video (Ballé et al., 2017; Agustsson et al., 2020), neural compression for EO remains fragmented, with no available large-scale pretrained codecs for multispectral imagery, and temporal redundancy in satellite time series still largely unexplored.

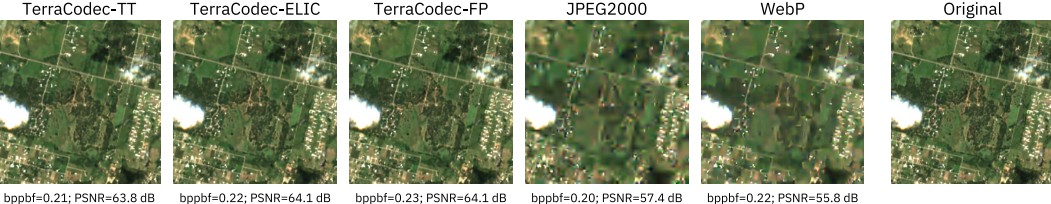

| TerraCodec-TT | TerraCodec-ELIC | TerraCodec-FP | JPEG2000 | WebP | Original |
|---|---|---|---|---|---|
| bppbf=0.21; PSNR=63.8 dB | bppbf=0.22; PSNR=64.1 dB | bppbf=0.23; PSNR=64.1 dB | bppbf=0.20; PSNR=57.4 dB | bppbf=0.22; PSNR=55.8 dB | |

Figure 1: Varying reconstruction quality at a similar compression rate.

We address these gaps with *TerraCodec*, a family of neural codecs tailored to multispectral EO and pretrained on Sentinel-2 data. TerraCodec includes efficient image-based models for multispectral inputs: a lightweight Factorized Prior variant (TEC-FP) and an ELIC-based variant (TEC-ELIC) for optimal rate-distortion. We further introduce a Temporal Transformer (TEC-TT) that captures

long-range dependencies across time without relying on hand-crafted motion priors. As shown in Figure 1, these models offer a significantly better reconstruction than standard codecs at the same compression rate. At equal image quality, they reduce storage size by up to an order of magnitude, with TEC-TT providing further reduction on long time series by exploiting temporal structure.

Our main contributions are: (1) **TerraCodec**, a suite of Sentinel-2 pretrained multispectral and multi-temporal codecs that enable evaluating the benefits of temporal modelling for EO compression and achieve superior rate–distortion performance over classical codecs; (2) **Latent Repacking**, a method to train variable-rate neural codecs which we demonstrate with our FlexTEC model; and (3) **downstream evaluations**, demonstrating the utility of compression models for downstream tasks and zero-shot cloud inpainting. We release code and pretrained weights under a permissive license to support future research and adoption.

## 2 RELATED WORK

**Foundations.** Shannon's source coding theorem bounds lossless compression by the source entropy; practical schemes such as Huffman and arithmetic coding approach this limit (Shannon, 1948; Huffman, 1952; Rissanen & Langdon, 1979). Lossy compression, in contrast, reduces storage requirements by discarding information. The rate–distortion function characterizes the minimum bitrate for a given distortion, formalizing the trade-off between rate and fidelity (Shannon, 1948). These principles underpin transform coding, which applies DCT or wavelets prior to quantization and entropy coding, forming the basis of standards like JPEG, JPEG2000, and HEVC (x265) (Ahmed et al., 1974; Daubechies, 1992; Wallace, 1991; Taubman & Marcellin, 2002; Sullivan et al., 2012).

**Neural compression.** Learned codecs replace hand-crafted transforms with autoencoders trained end-to-end under a rate–distortion loss (Ballé et al., 2017; Theis et al., 2017). Inputs are mapped to latents, quantized, and entropy-coded under a learned prior. Recent work extends this beyond convolutional autoencoders, exploring transformer backbones (Zhu et al., 2022; Li et al., 2024a) generative decoders (Yang & Mandt, 2023; Li et al., 2025) and latent diffusion models (Zhou et al., 2025), as well as richer perceptual and adversarial objectives (Blau & Michaeli, 2019; Mentzer et al., 2020). A key performance trade-off however is governed by the entropy model. Fully factorized priors offer efficiency (Ballé et al., 2017); hyperpriors (Ballé et al., 2018) introduce side information to capture spatially varying scales; autoregressive priors (Minnen et al., 2018) exploit local context at the cost of sequential decoding; and models such as ELIC (Cheng et al., 2020; He et al., 2022) additionally utilize efficient, parallel space–channel context. However, these image codecs are limited to a single rate–distortion setting per checkpoint. In contrast, flexible-rate models are using approaches such as conditioning on the rate parameter (Choi et al., 2019), spatially adaptive quality maps (Song et al., 2021; Tong et al., 2023), and hierarchical VAEs with quantization-aware priors (Duan et al., 2023).

Beyond images, learned video compression targets temporal redundancy across frames (Agustsson et al., 2020; Li et al., 2023; 2024b). While classical approaches rely on motion estimation and compensation, transformer-based models remove such priors and model temporal dependencies directly in latent space. The Video Compression Transformer (VCT) (Mentzer et al., 2022) follows this design, encoding frames independently and using a temporal transformer to predict latents from past context, making it better suited to settings with limited or irregular motion.

A complementary line of research investigates Implicit Neural Representations (INRs), which fit a small network to each image or video and store the signal in its weights, achieving strong rate–distortion performance but requiring per-sample optimization (Kim et al., 2024; Gao et al., 2025; Zhang et al., 2024a; Li et al., 2024c).

**Earth Observation data.** Most neural compression targets natural imagery, whereas EO includes multispectral bands, higher bit depth, and long temporal horizons. Compression must preserve spectral and structural cues relevant for downstream analysis (Gomes et al., 2025), aligning with the broader paradigm of task-oriented compression (Torfason et al., 2018; Singh et al., 2020). Operational EO pipelines typically rely on the JPEG2000 and CCSDS standards (Yeh et al., 2005; CCS, 2012; 2017) for their robustness and low complexity. Learned models have been explored for optical and SAR (Maharjan & Li, 2023; Di et al., 2022) images, with a focus on reducing on-board complexity (Alves de Oliveira et al., 2021) and spectral grouping (i Verdú et al., 2023). Other

works exploit spatial–spectral encoders and mixed hyperpriors to capture redundancy (Kong et al., 2021b;a; Cao et al., 2022; Xiang & Liang, 2023; Fu & Du, 2023; Gao et al., 2023). Recent generative approaches focus on low-bitrate RGB imagery using diffusion models (Ye et al., 2025; Zhang et al., 2024b), and INR–based methods have been explored for multispectral data (Cho et al., 2024). Despite this progress, EO compression research has predominantly focused on single-image settings and often relies on RGB or small-scale datasets, with limited exploration of temporal modeling. To our knowledge, no pretrained models are publicly available for the widely used Sentinel–2 imagery. TerraCodec aims to fill these gaps and offers multispectral neural codecs, a temporal transformer model to capture long-range dependencies, and single-checkpoint, flexible-rate compression.

## 3 METHODOLOGY

We begin with an overview of our EO compression approach, then detail the architectures of the TerraCodec models 3.1, and finally introduce Latent Repacking for flexible-rate models 3.2.

We study *lossy compression* of multispectral, multi-temporal EO imagery. An EO sequence is a set of images $\mathbf{x}_i \in \mathbb{R}^{H \times W \times C}$, each of size $H \times W$ with $C$ spectral bands. While EO sensors range from a single panchromatic channel to finely sliced hyperspectral imagers, we focus on Sentinel–2 L2A with $C{=}12$ optical bands in the visible and near infrared, saved as 16-bit radiometry.

A learned codec encodes a frame via an analysis transform $\mathbf{y}_i = g_a(\mathbf{x}_i)$, then quantizes and entropy-codes the latents $\hat{\mathbf{y}}_i = \mathcal{Q}(\mathbf{y}_i)$. The synthesis transform reconstructs the frame, $\hat{\mathbf{x}}_i = g_s(\hat{\mathbf{y}}_i)$. Compression relies on an entropy model $q_\phi(\hat{\mathbf{y}})$ that approximates the unknown latent distribution $p(\hat{\mathbf{y}})$, so arithmetic coding spends, in expectation, the cross-entropy $R \approx \mathbb{E}_{\hat{\mathbf{y}}_i \sim p}\big[ -\log_2 q_\phi(\hat{\mathbf{y}}_i)\big]$. We train $g_a$, $g_s$, and $q_\phi$ end-to-end with the rate–distortion loss $\mathcal{L} = R + \lambda D$, where $D$ denotes reconstruction error between $\mathbf{x}_i$ and $\hat{\mathbf{x}}_i$, which in our case is measured as MSE in standardized space. As $q_\phi$ better approximates $p$, the achieved rate $R$ approaches the entropy $H(p) = \mathbb{E}_p[-\log_2 p]$. Training thereby learns the entropy model while also shaping the latent space to be more predictable under $q_\phi$.

**EO-specific choices.** Our codecs adopt three EO-specific design choices: (i) native support for 12-band, 16-bit inputs; (ii) pretraining on a large-scale global EO dataset; and (iii) per-band standardization rather than global normalization, to stabilize training and preserve band-specific statistics for downstream tasks. Following prior literature (Ballé et al., 2017; Mentzer et al., 2022), we adopt CNN-based encoders for intra-frame compression due to their efficiency and low latency.

### 3.1 TERRACODEC

We introduce *TerraCodec*, a family of learned codecs for EO, including two image codecs: a lightweight factorized prior (TEC-FP), a stronger space–channel context model (TEC-ELIC), and a temporal transformer (TEC-TT), with a flexible-rate variant (FlexTEC).

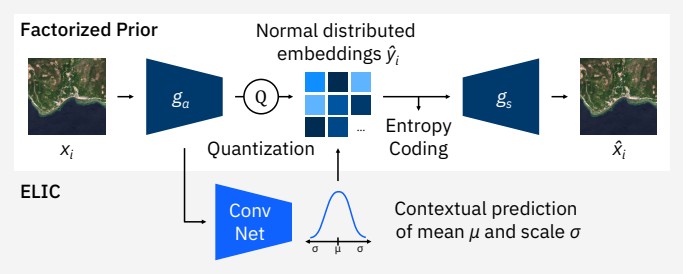

Figure 2: TerraCodec image codecs. Factorized Prior (Ballé et al., 2017) uses a fully factorized prior without context, and assumes a zero-centered normal distribution for the latents. ELIC (He et al., 2022) augments a hyperprior with spatial and channel context to predict per-latent mean/scale.

**Factorized Prior (TEC-FP).** TEC-FP is a Factorized Prior model (Ballé et al., 2017), our most basic TEC image codec. It employs a fully factorized entropy model, where each element of the quantized latent $\hat{\mathbf{y}}$ is modeled independently by $q_\phi(\hat{y}_j)$, without side information or context. It is illustrated

in the upper part of Figure 2. This yields fast, parallel entropy coding, but limited expressiveness compared to hyperprior- or context-based models.

**Efficient Learned Image Compression (TEC-ELIC).** TEC-ELIC instantiates ELIC's space–channel context entropy model (He et al., 2022) for EO inputs. The encoder/decoder networks include residual bottleneck and attention blocks, increasing representational capacity. The entropy model predicts per-latent mean and scale from (i) spatial context via checkerboard convolutions, (ii) channel context from previously decoded latent groups, and (iii) side information from a hyperprior, improving rate–distortion performance at the cost of higher complexity. Figure 2 illustrates the hyperprior-based context model in simplified form.

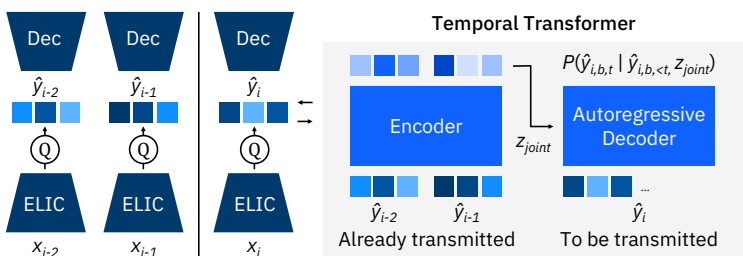

Figure 3: Architecture of the TerraCodec-TT model following Mentzer et al. (2022). Each input image is first encoded into latents by an ELIC image encoder. The per-image latents are tokenized, and a temporal transformer models these tokens autoregressively, predicting the mean and scale parameters for the current frame's tokens based on past latents.

**Temporal Transformer (TEC-TT).** TEC-TT builds on the VCT architecture (Mentzer et al., 2022). We train a transformer to model temporal dependencies of seasonal EO data in latent space, predicting the current frame's latent distribution from past context. Each frame $\mathbf{x}_i$ is encoded to latents $\mathbf{y}_i$ and quantized. We partition the current latent into $B$ non-overlapping spatial blocks $\{\hat{\mathbf{y}}_{i,b}\}_{b=1}^{B}$ and the two past latents into overlapping context blocks to increase the the transformer's receptive field. Each block $b$ is flattened into a sequence of $T$ tokens $\{\hat{y}_{i,b,t}\}_{t=1}^{T}$ of channel width $d_{\text{lat}}$. A temporal encoder aggregates the two previous frames into a joint context embedding $z_{\text{joint}} = E(\hat{\mathbf{y}}_{i-2}, \hat{\mathbf{y}}_{i-1})$. As shown in Figure 3, within each current block, a masked autoregressive transformer predicts token-wise prior parameters conditioned on already decoded tokens and $z_{\text{joint}}$ following Eq. 1.

$$p(\hat{\mathbf{y}}_{i,b,t} \mid \hat{\mathbf{y}}_{i,b,<t}, z_{\text{joint}}) = \prod_{d=1}^{d_{\text{lat}}} \mathcal{N}\big(\hat{y}_{i,b,t}^{(d)}; \mu_{i,b,t}^{(d)}, (\sigma_{i,b,t}^{(d)})^2\big) \tag{1}$$

We assume conditional independence across blocks given the context, allowing parallel probability estimation during encoding and parallel block decoding. Causal masking prevents attention to undecoded tokens; see Appendix B.3 for details. TEC-TT uses the same CNN analysis–synthesis transforms as TEC-ELIC. Unlike the original VCT, it is trained end-to-end on the rate–distortion objective without image pretraining, using a $\lambda$-schedule that emphasizes low-rate regimes early. We further adapt TEC-TT for flexible-rate scaling, introducing the FlexTEC variant in the next section.

## 3.2 LATENT REPACKING FOR FLEXIBLE-RATE MODELS

Most neural codecs are trained for a fixed rate–distortion tradeoff. This makes deployment inflexible since achieving different bitrates requires retraining separate models. Our goal is to support variable rates at inference. We introduce Latent Repacking, which redistributes latent channels across tokens, and apply token masking with dynamic rate scaling during training so tokens learn an information-based ordering. Early tokens capture global structure, later ones refine detail. Truncating tokens then lowers bitrate while preserving global content. We demonstrate this by adapting TEC-TT, where strong priors allow missing tokens to be predicted, making it well-suited for Latent Repacking.

**From spatial tokens to channel slices.** A standard transformer codec represents an image block with $T$ spatial tokens, each spanning the full latent dimension $d_{\text{lat}}$. Dropping tokens at inference discards entire regions, causing severe artifacts (see App. F.2.1). Instead, we aim for early tokens to encode information that is *globally useful* across the scene.

Think of the latent block as a 3D tensor $A \in \mathbb{R}^{H \times W \times d_{\text{lat}}}$, with $H \times W = T$ tokens and latent dimension $d_{\text{lat}}$. In the standard layout, each token $t$ is a spatial patch $(h, w)$ containing all $d_{\text{lat}}$ channels at that location. Latent repacking instead slices the channel axis into $T$ groups of width $k = \frac{d_{\text{lat}}}{T}$, and redefines tokens so that each spans the full scene but only $k$ channels.

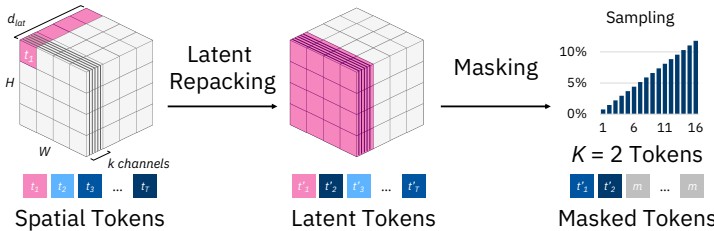

Figure 4: Latent Repacking converts $T$ spatial tokens $(W \cdot H)$ into channel-slice tokens so each token carries scene-wide content. During training, we sample a token budget $K$, mask the rest using a learned token $m$, and scale the rate. During inference, the user can pick the compression level $K$.

**Formal definition.** We define $T$ new tokens $\{t'_1, \ldots, t'_T\}$, each formed by a slice of $k$ channels across all spatial positions. Concretely, the $u$-th repacked token is

$$t'_u = A[:, :, (u-1) \cdot k : u \cdot k] \ \in \ \mathbb{R}^{H \times W \times k}. \tag{2}$$

In other words, $t'_u$ is the $u$-th slice of $k$ consecutive channels of $A$, spanning the full spatial field. The procedure is reversible; reapplying the slicing and repacking restores the original layout. After repacking, keeping the first $K$ tokens corresponds to $A[:, :, 0 : K \cdot k]$, i.e. the first $K \cdot k$ latent channels at every spatial location.

**Masked training.** In order for a flexible-rate model to learn varying rate settings, we mask the repacked tokens during training. Therefore, we sample a token budget $K \in \{1, \ldots, T\}$ and mask the last $T - K$ tokens by replacing them with a learned mask token $m$. Let $M_u \in \{0, 1\}$ indicate masking ($M_u = 1$ if token $t'_u$ is kept). Assuming an autoregressive model with additional context $c$, the rate is computed only over unmasked tokens as shown in Eq. 3.

$$R(M) \ = \ \sum_u M_u \circ \left[ -\log_2 \, q_\phi(t'_u \,|\, t'_{<u}, c) \right] \tag{3}$$

To keep later tokens informative, we use *dynamic rate scaling*: when fewer tokens are kept, their rate loss weight is upweighted. It prevents information from collapsing into the first tokens and encourages useful content across all tokens. Budgets $K$ are sampled more frequently at higher values (as in Bachmann et al. (2025)), ensuring that *all* tokens are trained while the model also learns to operate across a range of rates.

We apply the approach on TEC-TT and introduce the **Flexible-Rate TerraCodec (FlexTEC)** model. The model applies Latent Repacking and masking inside the temporal transformer after image-wise compression and restores the original layout before image decoding (details in App. C).

## 4 EXPERIMENTAL SETUP

This section describes the data and pretraining (4.1), evaluation and baselines (4.2), and downstream tasks (4.3) used to train and assess TerraCodec.

### 4.1 PRETRAINING

All TEC models are pretrained on SSL4EO-S12 v1.1 (Blumenstiel et al., 2025; Wang et al., 2023), a large-scale Sentinel–2 corpus with 244k globally distributed locations and four seasonal snapshots per location. Each L2A sample consists of $264 \times 264$ pixels at 10 m resolution; we crop to $256 \times 256$ pixels (random crops for training, center crops for evaluation) to ensure uniform size. Bands at 20 m and 60 m are upsampled to 10 m with nearest-neighbor interpolation for spatial alignment. We

follow the official spatial split into training and validation sets. To stabilize multispectral training, each input band $b$ is standardized by its dataset mean $\mu_b$ and standard deviation $\sigma_b$. Losses are computed in this standardized space, which balances gradient magnitudes across channels and avoids overfitting to high-variance bands.

**Image codecs.** TEC-FP and TEC-ELIC are optimized with Adam ($lr = 10^{-4}$), using an auxiliary learning rate of $5 \cdot 10^{-3}$ for the entropy bottleneck, gradient clipping at 1.0, and mixed precision. We employ a cosine learning-rate schedule with 5% warmup and $\eta_{\min} = 10^{-5}$. Models are trained for 100 epochs with batch size 64 on a single NVIDIA A100 GPU, requiring 20–25 hours. A temporal index is randomly sampled for each sample in every epoch, so that one epoch covers one quarter of the dataset. We sweep five $\lambda$ values to span low- to high-bitrate regimes.

**Temporal codec.** TEC-TT is trained with a temporal context of two past frames for 300k steps with a global batch size of 24 on four NVIDIA A100 GPUs, requiring about 70 hours. We optimize with AdamW ($lr = 10^{-4}$, weight decay $= 10^{-2}$) and employ half-cosine decay schedule to $\eta_{\min} = 10^{-6}$ with 15% warmup steps. We sweep six $\lambda$ values for varying rate–distortion settings. For low-rate settings ($\lambda \leq 5.0$), we scale $\lambda$ by 10 during the first 15% of training before annealing back to the target value. Further pretraining details, including FlexTEC, are given in Appendix B.

## 4.2 EVALUATION

**Baselines.** We compare TerraCodec to widely used classical codecs: JPEG (Wallace, 1991), JPEG2000 (Taubman & Marcellin, 2002), WebP (Zern et al., 2024), and HEVC (x265) (Sullivan et al., 2012), using the highest bit support for each codec. For JPEG and WebP we use the Pillow library; for JPEG2000 we use Glymur; and for HEVC we use the ffmpeg x265 implementation with the *medium* preset tuned for PSNR. Since these codecs are limited to three-channel RGB, we apply them per band, encoding each spectral channel as an independent grayscale image, following prior work in EO compression (Radosavljevic et al., 2020; Grassa et al., 2022).

**Metrics.** Compression rate is reported as *bits-per-pixel-band-frame* (bppbf), which normalizes by spatial resolution, number of spectral bands, and sequence length. Unlike the standard *bits-per-pixel* (bpp) used for three-channel RGB images, bppbf extends to arbitrary channel counts and sequence lengths. Distortion is quantified using PSNR, SSIM (Wang et al., 2004), MS-SSIM (Wang et al., 2003), and MSE in the destandardized 16-bit reflectance space. To ensure equal contribution of all spectral channels, metrics are computed per band and averaged across channels, avoiding bias toward bands with higher variance.

## 4.3 DOWNSTREAM TASKS

**Cloud inpainting.** TEC-TT captures spatiotemporal priors that can be applied beyond compression. We demonstrate this with zero-shot cloud removal on the AllClear benchmark (Zhou et al., 2024). Each sample consists of three cloudy observations with masks and one cloud-free target. To apply TEC-TT, we use the two least cloudy images as a temporal context and extract all cloud-free patches from the least cloudy image as input $x_0$. Patches in $x_0$ that are covered by clouds are predicted by TEC-TT. We report PSNR and other metrics, comparing against the official AllClear baselines. Following the benchmark, metrics are computed in auto mode across all bands, not per-band.

**Downstream models on compressed data.** To evaluate the effect of compression on downstream analysis, we benchmark downstream task models on compressed–reconstructed versus uncompressed inputs. Following standard EO practice, we finetune task-specific encoder–decoder models on either reconstructed or original inputs. We evaluate on Sen1Floods11 (Bonafilia et al., 2020), consisting of $512^2$ patches from 11 flood events, with binary segmentation masks. The original dataset includes Sentinel–2 L1C imagery. We, therefore, redownload the L2A version for TEC-FP. For patchwise multi-label land cover classification, we use reBEN-7k (Marti-Escofet et al., 2025), which spans eight countries and 19 semantic labels. Images have $120^2$ pixels; we apply *reflect* padding to match the model input sizes. We compress all inputs at three different operating points using TEC-FP, then fine-tune pretrained models on the reconstructed data. We employ Prithvi 2.0 100M (Szwarcman et al., 2025) and TerraMind base (Jakubik et al., 2025) backbones, with a UNet decoder for segmentation and a linear head for classification. All models are trained for 100 epochs with AdamW ( $lr = 5 * 10^{-5}$) and a *reduce-on-plateau* scheduler.

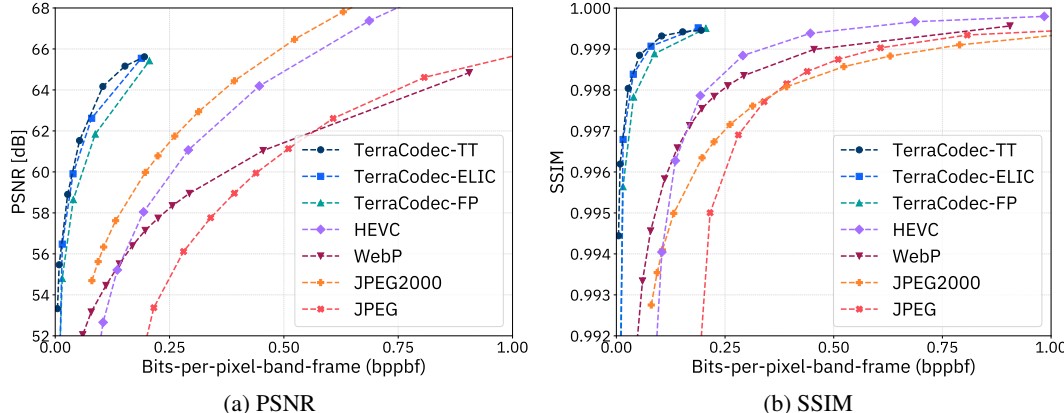

(a) PSNR  (b) SSIM

Figure 5: Rate–distortion curves for PSNR (↑) and SSIM (↑) on SSL4EO-S12 v1.1 validation sequences. TerraCodec models consistently outperform standard codecs, with TEC-TT achieving the best overall performance by exploiting temporal context.

## 5 EXPERIMENTS

We evaluate TerraCodec in terms of rate–distortion (Section 5.1), flexible-rate compression (Section 5.2), and utility for downstream EO tasks (Section 5.3).

### 5.1 RATE–DISTORTION

Figure 5 reports RD curves (bppbf vs. PSNR and SSIM) on the SSL4EO-S12 v1.1 validation set. TerraCodec consistently outperforms classical image and video codecs in both metrics. The lightweight TEC-FP achieves up to $5\times$ lower rate than the best image codec WebP at an equal SSIM of 0.999, while TEC-ELIC and TEC-TT provide further compression gains. Qualitative reconstructions (App. F.1) highlight that TerraCodec preserves finer structures and details compared to classical codecs. JPEG2000 achieves competitive PSNR with only 3x lower compression rate at similar distortion – surpassing HEVC – but performs poorly in SSIM, especially at high quality. This arises from its tendency to preserve pixel averages (favored by PSNR/MSE) while oversmoothing textures and edges that SSIM is sensitive to. In contrast, TerraCodec models maintain strong performance across both metrics, demonstrating efficient compression without loss of high-frequency details.

TEC-TT improves over the neural image models, although its margin over TEC-ELIC is smaller than the typical video–image gap. While EO sequences differ from natural video, being sampled at daily to seasonal rather than sub-second intervals, the short 4-frame validation setup further departs from typical video settings.

This limits temporal gains since half of the frames are *bootstrap* frames, which lack full previous context, compared to *P-frames*, which are predicted from two past frames. To better understand these effects, we study the role of temporal conditioning and report the RD performance on *P-frames only* in Figure 6. Without context, TEC-TT reduces to an image codec (labeled as *image only*). For a medium setting ($\lambda = 5$), conditioning on one past frame improves compression by 13.6% compared to no context. On P-frames with two previous images, TEC-TT achieves a 22.6% rate reduction at equal PSNR, showing that longer EO sequences naturally yield greater efficiency as bootstrap frames are amortized.

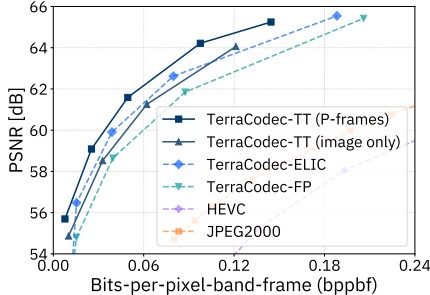

Figure 6: Rate–distortion curves for PSNR (↑) on all evaluation P-frames.

## 5.2 FLEXIBLE RATE–DISTORTION

Our TEC-TT-based FlexTEC model uses Latent Repacking to provide flexible rate-compression from a single checkpoint by transmitting a variable subset of latents and inferring the remainder from the model prior. Figure 7 compares FlexTEC against our fixed-rate models and standard codecs on P-frame compression. While fixed-rate TEC-TT models serve as an upper bound—each being optimally fitted for one specific rate–distortion—they require several separately trained models. In contrast, FlexTEC provides several user-controlled rate settings and performs close to or better than TEC-FP, depending on the setting. Further analysis in Appendix E.3 shows that FlexTEC encodes significant information in bootstrap frames, leading to similar bitrates independent of the token budget. The model then uses this information in the following P-frames to provide efficient rate–distortion settings. Figure 8 provides qualitative examples for different token budgets.

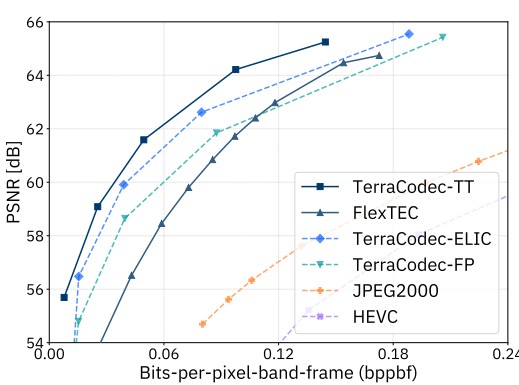

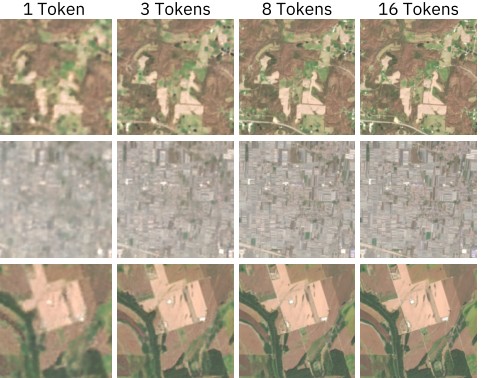

Figure 7: RD curves for PSNR (↑) on P-frames. FlexTEC performs close to fixed-rate models and significantly better than standard codecs.

Figure 8: FlexTEC reconstructions with different token budgets. Early tokens capture coarse structures, while later tokens refine details.

## 5.3 DOWNSTREAM TASKS

We study the usability of TEC models beyond rate–distortion compression by examining how their learned priors (*model beliefs*) can be leveraged for zero-shot prediction and how compression affects downstream EO applications.

**Model beliefs.** Neural codecs rely on learned priors to estimate latent distributions for entropy coding. By decoding these priors into the image space, we obtain predictions that expose the model's implicit knowledge. Figure 9 shows qualitative examples, with additional results in Appendix F.2.2. We compare priors under three conditions: (a) no information from the current frame (only past context, TEC-TT), (b) partial information from the current latent (past frames and a subset of tokens for TEC-TT; subsets of channel groups for TEC-ELIC), and (c) full information of the latent. TEC-FP, lacking a hyperprior, cannot adapt to the latent at hand. Results show that TEC-TT already produces plausible forecasts from past frames alone and improves its beliefs when partial information becomes available, yielding the most refined predictions.

**Cloud inpainting.** Building on TEC-TT's latent predictions, we evaluate zero-shot cloud removal on the AllClear benchmark (Zhou et al., 2024). TEC-TT is applied without task-specific training (see Sec. 4.3 and Appendix G), with results summarized in Table 1 and qualitative examples in Figure 10. In addition to the full test set, we report performance on subsets ranked by cloud coverage.

The benchmark compares against heuristic baselines (LeastCloudy, Mosaicing) and prior zero-shot neural approaches. TEC-TT outperforms all heuristic methods and prior zero-shot neural models on the full test set. The subset analysis, focusing on the most challenging samples in terms of cloud coverage, further underscores the benefit of temporal priors: heuristic approaches perform adequately in low-cloud cases, but break down under heavy cloud coverage, while TEC-TT maintains high

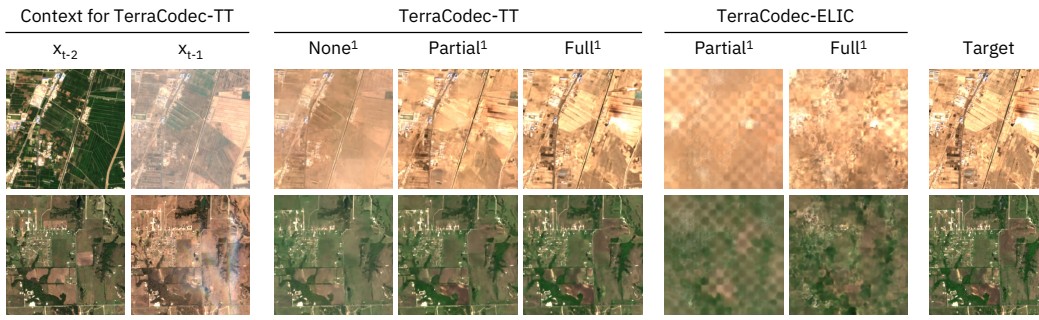

| Context for TerraCodec-TT | | TerraCodec-TT | | | TerraCodec-ELIC | | |
|---|---|---|---|---|---|---|---|
| $x_{t-2}$ | $x_{t-1}$ | None[1] | Partial[1] | Full[1] | Partial[1] | Full[1] | Target |

[1] Model knowledge of current frame (target)

Figure 9: Model beliefs obtained by decoding learned priors into the image space. TEC-TT is shown with 0 (past context only), 5 and 16 tokens. TEC-ELIC uses limited and full channel-group context.

PSNR. The hardest 10% of samples correspond to an average cloud coverage of 99%, yet TEC-TT still produces reasonable predictions. Overall, the results demonstrate that the temporal modeling in TEC-TT not only improves compression but also transfers to challenging forecasting tasks.

Table 1: Test PSNR on AllClear across difficulty subsets (by average cloudiness). PSNR computed across all bands following AllClear.

| **Model** | **10%** | **20%** | **50%** | **100%** |
|---|---|---|---|---|
| *Baseline heuristics* | | | | |
| LeastCloudy | 11.07 | 14.08 | 24.82 | 30.61 |
| Mosaicing | 16.55 | 16.70 | 23.73 | 29.82 |
| *Pre-trained models (zero-shot setting)* | | | | |
| CTGAN | 25.58 | **26.60** | 27.59 | 27.79 |
| DiffCR | 24.55 | 25.13 | 25.50 | 25.21 |
| PMAA | 24.82 | 25.06 | 25.02 | 24.32 |
| U-TILISE | 13.20 | 14.95 | 18.33 | 24.67 |
| UnCRtainTS | **26.42** | 26.50 | 27.97 | 29.01 |
| **TerraCodec-TT** | 25.97 | 26.59 | **30.38** | **32.86** |

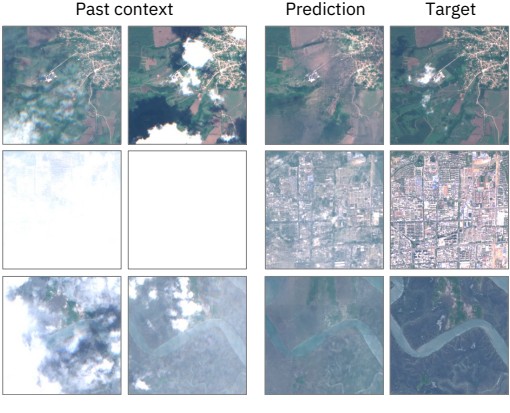

Figure 10: Cloud inpainting examples with TEC-TT on the AllClear benchmark.

**Downstream models.** In EO, downstream task models are typically trained on uncompressed data to avoid information loss, requiring to transmit and store large volumes of raw data. We investigate the impact of training and evaluating on data compressed with TerraCodec-FP. Table 2 reports image analysis results on reBEN-7k and Sen1Floods11. Models fine-tuned after moderate compression lead to a performance drop of <1.0pp across all metrics, while reducing data size by up to $380\times$. At high compression, we observe more pronounced degradations: the F1 score on reBEN-7k decreases by 3.4pp with TerraMind, and the $IoU_{Flood}$ on Sen1Floods11 drops by 2pp with both models. These results suggest that moderate compression can be employed without substantial impact on downstream analysis, whereas higher compression levels entail some performance trade-off.

## 6 CONCLUSION

We introduce and release TerraCodec, a family of learned compression models for Earth observation, pretrained on Sentinel-2 multispectral time series data. Our models outperform classical image and video codecs in rate–distortion, achieving up to an order-of-magnitude reduction at equal quality. Latent Repacking further enables flexible-rate transformer models from a single checkpoint, as demonstrated by FlexTEC. Downstream evaluations show that moderate compression preserves analysis performance, while zero-shot cloud inpainting highlights the strengths of our temporal transformer TEC-TT beyond compression. Overall, TerraCodec establishes a foundation for explor-

Table 2: Test performance on reBEN-7k and Sen1Floods11 (↑) when training on compressed inputs (TEC-FP). Numbers in parentheses show the change relative to training on uncompressed data. Performance remains stable at low and mid rates, with clearer degradation for high compression.

| Task Model | Compression | reBEN-7k | | Sen1Floods11 | |
|---|---|---|---|---|---|
| | | **Accuracy** | **F1** | **mIoU** | **IoU**$_{Flood}$ |
| TerraMind base | Original data | 88.76 | 61.99 | 87.77 | 78.75 |
| | 170× | 89.05 (+0.29) | 63.24 (+1.25) | 87.31 (-0.46) | 78.02 (-0.73) |
| | 380× | 88.82 (+0.06) | 60.97 (-1.02) | 87.27 (-0.50) | 77.97 (-0.78) |
| | 940× | 87.80 (-0.96) | 58.60 (-3.39) | 86.76 (-1.01) | 77.06 (-1.69) |
| Prithvi 2.0 100M TL | Original data | 87.93 | 59.23 | 87.27 | 77.92 |
| | 170× | 87.42 (-0.51) | 59.14 (-0.09) | 87.06 (-0.21) | 77.53 (-0.39) |
| | 380× | 87.06 (-0.87) | 60.15 (+0.92) | 86.61 (-0.66) | 76.86 (-1.06) |
| | 940× | 86.86 (-1.07) | 58.28 (-0.95) | 85.96 (-1.31) | 75.81 (-2.11) |

ing the benefits of temporal modelling in EO compression and for advancing multispectral learned compression in Earth observation.

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

# A APPENDIX

## A.1 LIMITATIONS

While TerraCodec is pretrained on a large and diverse global dataset, the current models are limited to Sentinel-2 L2A imagery. The underlying architectures and training pipeline are sensor-agnostic, but transferring to other sensors requires finetuning or retraining. From a compression perspective, this reflects a common trade-off: sensor-specific codecs typically achieve higher compression efficiency by modeling the statistics and dynamic range of a particular instrument, whereas sensor-agnostic models offer broader applicability. We see cross-sensor and multisensor pretraining as promising future directions, while also recognizing the practical value of sensor-specific codecs in operational EO pipelines. Our implementation, which will be open-sourced, prioritizes research clarity over optimized inference speed; for instance, efficiency techniques such as KV caching are not yet integrated. Incorporating such improvements could further enhance the models' practicality for real-world deployment.

Our temporal models operate with a two-frame context and are trained and evaluated on four-frame sequences. Using longer sequences would likely reveal additional temporal gains but would also incur substantially higher computational cost. Our p-frame ablation (Fig. 6) highlights the temporal benefits achievable under longer-sequence evaluation.

Finally, FlexTEC establishes a strong flexible-rate baseline, leveraging a latent inference mechanism in which rate control is achieved by dropping latent tokens and inferring the corresponding missing parts through the temporal latent model at decode time. While FlexTEC performs competitively across many settings, there remains room to further narrow the gap to fixed-rate models, particularly under very high compression ratios. Exploring hybrids with other variable-rate methods is a promising future direction.

## A.2 USE OF LLMs

We utilized large language models (LLMs) to refine text, improve readability, and assist with coding. All methods, technical content, experimental design, and analyses were developed by the authors.

# B TECHNICAL IMPLEMENTATION DETAILS

This section provides additional TerraCodec implementation details not included in the main paper.

## B.1 FRAMEWORK AND ENVIRONMENT

All models are implemented in PyTorch, using CompressAI (Bégaint et al., 2020) for core architectures and entropy bottlenecks, extended to support multispectral inputs, temporal samples, and model-belief analyses. The TEC-TT implementations are based on Muckley et al. (2021). Experiments are run on NVIDIA A100 GPUs with mixed precision training.

## B.2 IMAGE MODELS (TEC-FP, TEC-ELIC)

**TerraCodec-FP** follows the factorized prior baseline (Ballé et al., 2017), with $g_a$ and $g_s$ implemented as four strided $5 \times 5$ convolutions combined with GDN/IGDN nonlinearities.

**TerraCodec-ELIC** builds on the uneven channel-group entropy model (He et al., 2022), using the Chandelier reimplementation (Chandelier, 2023). Latents are divided into groups $[16, 16, 32, 64, M-128]$ and coded sequentially using SCCTX (space–channel context). Relative to He et al. (2022), we omit the preview head for efficiency. Despite the stronger context model, training time remains comparable to TEC-FP since the channel-group autoregression is parallelizable across spatial locations and implemented with efficient masked convolutions.

**Checkpoint settings.** Hyperparameters $N$ and $M$ denote the channel width of encoder/decoder layers and the latent bottleneck size, respectively. For both codecs, $N$ and $M$ are scaled slightly across the trained $\lambda$ values, as summarized in Table 3. All checkpoints are trained with identical optimization settings (see Sec. 4.1), varying only $N$, $M$, and the rate–distortion trade-off coefficient $\lambda$.

Table 3: Architectural specifications for TerraCodec-FP and TerraCodec-ELIC. $N$: main network channels; $M$: latent bottleneck channels.

| Family | Model ($\lambda$) | Analysis / Synthesis | Channels ($N/M$) |
|---|---|---|---|
| TerraCodec-FP | $\lambda = 0.5$ | Conv layers GDN / IGDN Downsampling ×16 | 128 / 128 |
| | $\lambda = 2$ | | 128 / 128 |
| | $\lambda = 10$ | | 128 / 128 |
| | $\lambda = 40$ | | 128 / 192 |
| | $\lambda = 200$ | | 192 / 320 |
| | $\lambda = 800$ | | 192 / 320 |
| TerraCodec-ELIC | $\lambda = 0.5$ | Conv layers Residual blocks Attention Downsampling ×16 | 128 / 192 |
| | $\lambda = 2$ | | 128 / 192 |
| | $\lambda = 10$ | | 128 / 192 |
| | $\lambda = 40$ | | 128 / 192 |
| | $\lambda = 200$ | | 320 / 320 |

**Temporal sampling.** For image models, we treat the pretraining data as an image dataset by sampling individual timesteps from the SSL4EO-S12 time series. During training, a single temporal index is randomly drawn for every sample in each epoch, such that one epoch covers one quarter of the temporal data. Over the 100 training epochs, this amounts to about 25 full passes through the complete dataset.

### B.3 TEMPORAL MODEL (TEC-TT)

We provide additional details on the TEC-TT architecture, including its tokenization strategy and temporal transformer design, following VCT (Mentzer et al., 2022).

The image encoder–decoder follows the TEC-ELIC backbone, composed of residual bottleneck and attention blocks with total downsampling ×16. For our $256 \times 256$ input crops, this yields a $16 \times 16$ latent grid with $M{=}192$ channels.

**Tokenization.** We tokenize the latent image representations using the scheme introduced in VCT. The latent grid $\hat{\mathbf{y}}_i \in \mathbb{R}^{H_\ell \times W_\ell \times d_{\text{lat}}}$ with $H_\ell = W_\ell = 16$ is divided into spatial blocks. The *current* frame is split into non-overlapping $4 \times 4$ blocks, each flattened into a sequence of $T{=}16$ tokens $\{\hat{\mathbf{y}}_{i,b,t}\}_{t=1}^{16}$. The *two past* frames are partitioned into overlapping $8 \times 8$ context blocks using reflect-padding so their grids align with the current frame, producing $T{=}64$ tokens per block.

Concretely, for block index $b \in \{1, \ldots, B\}$:

$$\text{current: } \{\hat{\mathbf{y}}_{i,b,t}\}_{t=1}^{16} \in \mathbb{R}^{16 \times d_{\text{model}}}, \qquad \text{past: } \{\hat{\mathbf{y}}_{i-1,b,t}\}_{t=1}^{64}, \quad \{\hat{\mathbf{y}}_{i-2,b,t}\}_{t=1}^{64}.$$

Each block forms an independent short token sequence, and all blocks are processed in parallel. All tokens are linearly projected to embeddings of width $d_{\text{tt}}{=}768$ before entering the temporal transformer.

**Temporal transformer stack.** The temporal model follows the VCT design and consists of two *separate encoders* $E_{\text{sep}}$ (one per past frame), a *joint encoder* $E_{\text{joint}}$ that fuses both contexts, and a *masked decoder* that autoregressively models the current block tokens conditioned on the fused context. We adopt the standard VCT specifications for the number of layers, heads, and embedding size in each transformer (see Table 4).

For token bootstrapping and inference, a learned start-of-sequence (SOS) token seeds masked decoding, while early frames without temporal context use a shared bias as a dummy prior (not entropy-coded).

**Training.** Training follows the standard uniform-noise quantization surrogate (Toderici et al., 2016; Minnen et al., 2018), while inference applies hard quantization and arithmetic coding. We train six TEC-TT variants at different rate–distortion trade-offs, controlled by $\lambda \in$

Table 4: TEC-TT transformer configuration. All blocks use GELU activations, pre-norm layers, and an MLP expansion factor of $4\times$. Dropout is disabled.

| Module | # Layers | # Heads | $d_{\text{model}}$ | # Tokens / patch |
|---|---|---|---|---|
| $E_{\text{sep}}$ (per past frame) | 6 | 16 | 768 | 64 |
| $E_{\text{joint}}$ (fusion) | 4 | 16 | 768 | $128^{\dagger}$ |
| Masked decoder (current) | 5 | 16 | 768 | 16 (causal) |
| Final heads $(\mu, \sigma)$ | $3\times$FC | – | 768 | out: $d_C{=}192$ |

$^{\dagger}$ Token count after concatenation of past-frame representations.

$\{0.4, 5.0, 20.0, 100.0, 300.0, 700.0\}$, where smaller values enforce higher compression and larger values prioritize reconstruction quality.

## C  LATENT REPACKING AND FLEXTEC

The main paper (Sec. 3.2) introduces *Latent Repacking*, which slices and reorders latent tokens such that early tokens encode global structure and later tokens refine local detail. Here, we provide additional intuition for introducing Latent Repacking and masked training, along with implementation details for FlexTEC.

**Scope and notation.**  FlexTEC is the flexible–rate variant of TEC–TT, obtained by integrating *Latent Repacking* and *masked training*. It uses the same analysis/synthesis (ELIC) backbone and temporal transformer stack as TEC–TT, with latent channel width $d_{\text{lat}}{=}192$. After tokenizing the current frame into $T{=}16$ tokens per patch, repacking groups channels into $T$ *channel-slice* tokens (Sec. 3.2). Consequently, FlexTEC exposes *16 discrete quality levels* via the token budget $K \in \{1, \ldots, 16\}$, applied consistently across all patches in an image and frames in a sequence. Each token carries $k = d_{\text{lat}}/T = 12$ channels shared across all spatial positions. For all rate–distortion (RD) visualizations in this paper, we report curves for budgets $K = \{1, 2, 3, 4, 5, 6, 7, 8, 12, 16\}$, spanning from the lowest-rate ($K{=}1$) to the highest-quality ($K{=}16$) operating points.

**Implementation differences vs. TEC–TT.**  FlexTEC is architecturally identical to TEC–TT except for: (i) the permutation that repacks tokens (Sec. 3.2); (ii) token masking with a learned mask token $m \in \mathbb{R}^{d_{\text{lat}}}$ used during training to replace dropped tokens; and (iii) the masked-rate objective with budget sampling. All other layers, dimensions, and hyperparameters remain unchanged.

FlexTEC is trained with the *same* hyperparameters as TEC–TT, but for 400k steps (vs. 300k for TEC–TT) to account for the task's added complexity. A single checkpoint trained at $\lambda{=}800$ (slightly higher than the TEC–TT maximum of 700) is used to cover the full bitrate range under masked training. Empirically, the $T/K$ scaling in the rate term shifts the effective operating point toward lower rates for the same $\lambda$, motivating this increase.

**Objective and inference.**  With mask $M$, the rate is computed on unmasked tokens and, to prevent information collapse into the earliest tokens, upweighted by $T/K$:

$$\mathcal{L} = \tfrac{T}{K}\, R(M) + \lambda D.$$

At test time we pick a budget $K$ (one of 16 levels), transmit only the first $K$ tokens, and *fill* dropped tokens with the transformer's predicted means $\mu$ before decoding. This yields graceful quality–rate scaling with a single checkpoint.

**Masking for variable-rate robustness.**  Repacking latents *alone* is insufficient: a model trained only with full-token inputs learns to rely on *all* tokens and collapses when some are dropped. We therefore train with *masked budgets*: sample $K \in \{1, \ldots, T\}$, replace the last $T{-}K$ tokens by a learned mask vector $m \in \mathbb{R}^{d_{lat}}$, and compute the rate only on unmasked tokens $R(M)$ (as defined in Eq. 3). For stability we use *teacher forcing*: masking applies only to the *current* frame's tokens for the rate term, while the temporal encoder always consumes the *real* quantized past latents $(\hat{\mathbf{y}}_{i-2}, \hat{\mathbf{y}}_{i-1})$. Inspired by Bachmann et al. (2025), budgets are drawn from a categorical distribution biased toward larger values, $\Pr(K{=}k) \propto k$ (i.e., the multiset $\{1, 2, 2, \ldots, T, \ldots, T\}$), which trains

the model frequently near high-rate operation while still exposing it to low-budget regimes. Masked tokens are replaced by a learnable per-channel vector $m \in \mathbb{R}^{d_{lat}}$ ($d_{lat}$=192), initialized uniformly in $[-1, 1]$, and shared across positions. This provides a stable placeholder during training while allowing the image decoder to learn how to interpret missing content.

While we keep the number of tokens $K$ fixed within each sequence, it could also be varied across time steps. One approach is to predict $K$ per sample, allowing the model to allocate tokens dynamically based on content complexity. We tested this by predicting $K$ from the joint latent representation $z_{\text{joint}}$ with a simple MLP, but found no improvement at inference, as $K$ appeared to be uncorrelated with perceptual complexity. We thus leave adaptive token rates as a future extension of Latent Repacking.

**Inference filling.** At inference, dropped tokens are *not transmitted*. By default, we fill them with the transformer's predicted means, i.e., $\hat{\mathbf{y}}_{i,b,t} \leftarrow \mu_{i,b,t}$ for frame $i$, latent block $b$, and token $t$, which leverages the learned temporal prior to improve reconstruction quality at a given bitrate. As a lighter alternative (reduced compute), we can instead substitute the learned mask vector $m$ for all dropped tokens.

# D EVALUATION DETAILS AND BASELINE METHODS

**Baseline quality settings.** We evaluate classical codecs across the following quality grids:

Table 5: Quality settings per codec used in RD evaluation.

| Codec | Quality settings |
|---|---|
| JPEG (Pillow) | 0, 1, 5, 15, 25, 35, 45, 55, 65, 75, 85, 95 |
| JPEG2000 (Glymur) | 0, 2, 5, 10, 15, 20, 25, 30, 40, 50, 60, 70, 80, 120, 150, 170, 200 |
| WebP (Pillow) | 0, 1, 5, 15, 25, 35, 45, 55, 65, 75, 85, 95 |
| HEVC/x265 (FFmpeg) | 5, 10, 15, 20, 25, 30, 35, 40, 45, 50 |

**Implementations and bit depth.** JPEG and WebP are executed via Pillow, supporting only 8-bit input. JPEG2000 is run per band using Glymur/OpenJPEG, which allows for 8- or 16-bit quantization and lossy compression via target ratios. HEVC encoding is performed with FFmpeg and x265, using raw 12-bit monochrome input (pixel format `yuv400p12le`), CRF= $Q$, `-preset medium`, and `-tune psnr`. Bitstream sizes are measured directly from encoded outputs to compute rate. All codecs operate *per band* (grayscale), and we report bits-per-pixel-band-frame (bppbf) by aggregating bitstream sizes across all bands and frames.

**Rate metric (bppbf).** We report rate as *bits per pixel–band–frame* (bppbf):

$$\text{bppbf} = \frac{\text{total bits}}{H \cdot W \cdot C \cdot T},$$

where $H \times W$ is the spatial resolution, $C$ the number of spectral bands, and $T$ the number of frames. Unlike standard bpp (suited to 3-channel RGB), bppbf normalizes across arbitrary channel counts and sequence lengths, enabling fair comparisons for multispectral time series.

# E QUANTITATIVE RESULTS

Across codecs, we report RD curves under multiple distortion metrics and normalizations, then isolate temporal effects (context window, P-frames) and flexible-rate behavior (masking ablation, amortization). This section complements the main text with analyses that clarify how metric choices and temporal conditioning impact conclusions.

## E.1 PSNR RANGE SENSITIVITY

In Fig. 11, we plot RD curves for all codecs using PSNR-per-band under different normalization ranges: PSNR 65k (full 16-bit), PSNR 10k (typical Sentinel-2 reflectance 0–10000), and PSNR auto

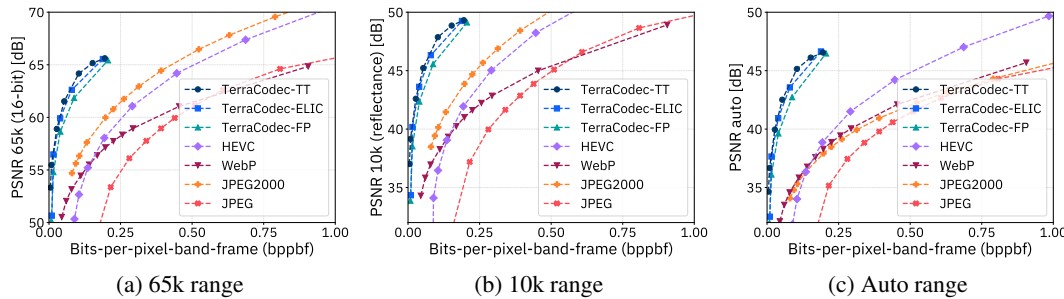

|  | (a) 65k range | (b) 10k range | (c) Auto range |

Figure 11: Comparison of the PSNR-per-band metric on different value ranges: 65k covering the full 16-bit range, 10k representing the typical 0–10000 reflectance range of Sentinel-2 data, and auto mode using min–max values of each band.

Table 6: Effect of inference context $c$ on TEC–TT (trained with 2-frame context, $\lambda$=5). We report bits-per-pixel–band–frame (bppbf, $\downarrow$) and PSNR-per-band (65k, $\uparrow$). Context $c \in \{0, 1, 1+1, 2\}$ denotes the number of conditioned frames. The P-frames row evaluates only the last two frames (full context) of each 4-frame sequence. Percent changes are relative to $c$=0.

| Setting | Context | bppbf $\downarrow$ | PSNR $\uparrow$ |
|---|---|---|---|
| No context (image codec) | 0 | 0.03274 | 58.522 |
| 1 previous frame | 1 | 0.02830 $(-13.6\%)$ | 58.761 $(+0.24\text{dB})$ |
| 1 previous frame (repeated) | 1+1 | 0.02785 $(-14.9\%)$ | 58.838 $(+0.32\text{dB})$ |
| 2 previous frames (all frames) | 2 | 0.02722 $(-16.9\%)$ | 58.902 $(+0.38\text{dB})$ |
| P-frames only (full context) | 2 | 0.02536 $(-22.6\%)$ | 59.085 $(+0.56\text{dB})$ |

(per-band min–max). We find that PSNR is sensitive to this choice. TerraCodec models (TEC–FP, TEC–ELIC, TEC–TT) remain comparatively stable across ranges, whereas the ranking of classical codecs shifts: JPEG2000 performs best under the PSNR 65k and PSNR 10k ranges, but in the *auto* setting at higher bitrates it is overtaken by WebP and x265.

### E.2 TEMPORAL CONDITIONING EFFECTS

We evaluate how temporal context influences fixed-rate TEC–TT models at inference. To isolate genuine temporal gains from the overhead of early bootstrap frames, we vary the number of available past frames and additionally analyze P-frame performance.

Table 6 extends the main paper's analysis of temporal conditioning. We additionally report results for ($c$=1+1) context, obtained by repeating the same past frame. This configuration yields only a marginal gain over using a single distinct frame (1.6% rate reduction), confirming that improvements stem from meaningful *temporal* information rather than simply longer input sequences. Restricting evaluation to P-frames (full context under $c$=2) further tightens the rate to 0.02536 bppbf at similar PSNR—an additional 6.8% reduction compared to all-frame results including bootstrap frames (0.02722) and 22.6% compared to no context. This quantifies the amortization effect discussed in the main paper and explains why four-frame sequences may understate the full temporal advantage.

Figure 12 shows corresponding RD curves evaluated on *P-frames only*—the last two frames in each four-frame sequence, consistent with TEC–TT's training context of two past frames. For non-temporal codecs, this selection simply aligns the evaluation set with TEC–TT. The performance gap between TEC–TT and image-only codecs (TEC–FP, TEC–ELIC) widens under this evaluation, reflecting the amortized cost of the initial bootstrap frames. When temporal conditioning is disabled and TEC–TT is run in "image mode," its performance closely follows TEC–ELIC, consistent with their shared analysis/synthesis backbone.

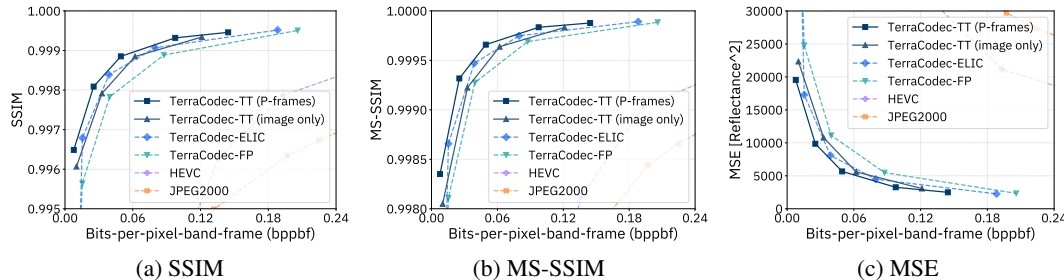

Figure 12: RD curves with additional metrics on the P-frame evaluation set.

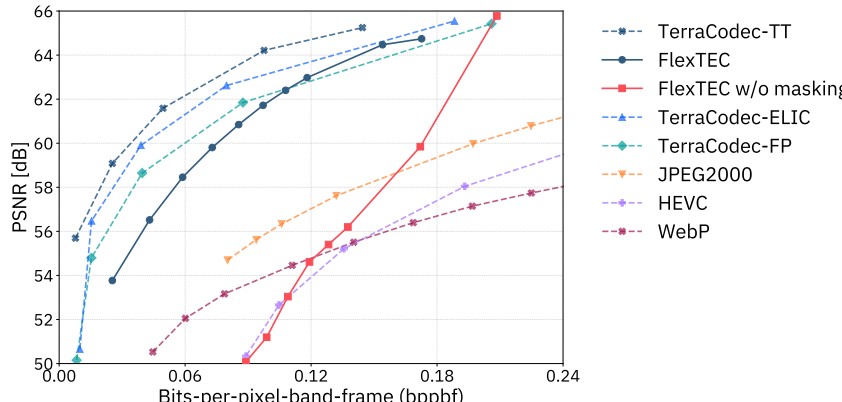

Figure 13: Masking ablation for FlexTEC (PSNR-per-band, 65k). We compare FlexTEC (Latent Repacking *with* masking) to a variant trained *without* masking under the same backbone and training setup. FlexTEC curves use token budgets $K = \{1, 2, 3, 4, 5, 6, 7, 8, 12, 16\}$.

### E.3 FLEXIBLE-RATE BEHAVIOR

We analyze how masked training impacts FlexTEC's variable-rate performance. We first ablate masking to verify its necessity, and then compare FlexTEC on P-frames versus all frames to quantify amortization effects.

Fig. 13 shows the effect of masking in flexible-rate training by comparing FlexTEC (Latent Repacking with masking) to an variant trained without masking. The latter deteriorates sharply when tokens are dropped at test time—its RD curve is unstable and substantially below FlexTEC—whereas FlexTEC degrades smoothly and remains roughly parallel to fixed-rate baselines. This confirms that masking is essential for stable variable-rate performance.

Fig. 14 compares FlexTEC on P-frames (last two frames, full context under $c=2$) versus all frames. The gap is notably larger than for fixed-rate TEC–TT, particularly at low rates. We hypothesize two compounding effects: (i) latent repacking with masked training encourages FlexTEC to concentrate scene-wide, high-utility content into the earliest tokens of *bootstrap* frames, lowering their cost relative to fixed-rate models; and (ii) for fully conditioned P-frames, the same curriculum distributes information more evenly across tokens, so truncation retains most of the essentials. Together, these effects yield stronger amortization when excluding bootstrap frames, with the benefit most pronounced in the high-compression regime.

## F QUALITATIVE EXAMPLES

This section complements the main paper with additional visual examples. We first compare reconstructions across all codecs on representative SSL4EO-S12 samples (Sec. F.1). We then examine *model beliefs* and forecasting behavior (Sec. F.2) for TEC-TT and FlexTEC. We assesss how dropping tokens impacts TEC-TT and the importance of token masking for Latent Repacking. We also

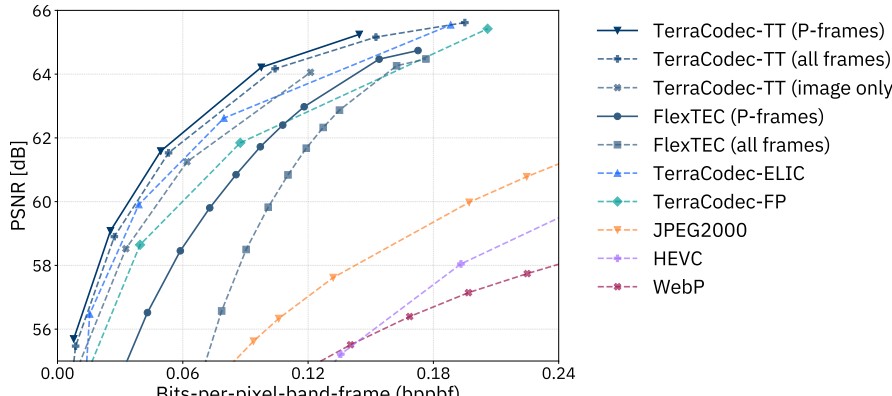

Figure 14: FlexTEC RD on P-frames vs. all frames (PSNR-per-band, 65k). The efficiency gains are much more pronounced for the flexible-rate model than for our fixed-rate TEC-TT models. FlexTEC curves use token budgets $K = \{1, 2, 3, 4, 5, 6, 7, 8, 12, 16\}$.

show how TEC–TT forecasts the next frame from past context, contrasting mean predictions with stochastic samples.

### F.1 GENERAL RECONSTRUCTIONS

We compare TerraCodec-TT (TEC-TT), TerraCodec-FP (TEC-FP), and classical codecs (JPEG2000, WebP) on SSL4EO-S12 v1.1 validation samples at matched rate $\approx 0.20$ bppbf. Each row in Fig. F.1) shows the original image and reconstructions from each codec, annotated with the average bppbf and PSNR-per-band (65k clipped) across the sequence.

### F.2 MODEL BELIEFS AND FORECASTING

We discuss the effect of token budget on different TT model versions and show the TEC-TT forecasts.

#### F.2.1 TOKEN BUDGET COMPARISON

We visualize token budget effects in Fig. 16 using two example sequences under an aggressive token limit, illustrating how models behave when later tokens are dropped. We compare TEC-TT, TEC-TT with Latent Repacking but no masking, and FlexTEC (with both). FlexTEC degrades smoothly and preserves scene-wide structure, whereas TEC-TT exhibits patch erasure and banding; the Latent Repacking w/o masking variant lies in between, confirming that masking with dynamic rate scaling is essential for stable variable-rate performance.

#### F.2.2 FORECASTS FROM PAST CONTEXT

We probe TEC–TT's *model beliefs* by predicting the current frame from past context only, using either the predicted mean or samples from the distribution (Fig. 17). The $\mu$-forecast reliably captures large-scale structure, while sampling from the full prior (mean and variance) expresses context-aware uncertainty, primarily reflecting cloud variability. This illustrates the learned distribution rather than a single point estimate. The conservative single-point forecast ($\mu$-forecast) produces a clear-sky prediction for the next frame, enabling TEC–TT to perform cloud removal as evaluated in AllClear.

## G ADDITIONAL DETAILS ON CLOUD INPAINTING

We provide extended results for the AllClear cloud inpainting benchmark (Zhou et al., 2024), complementing Sec. 4.3 and Table 1. TEC–TT is applied without task-specific fine-tuning, leveraging only its latent temporal predictions.

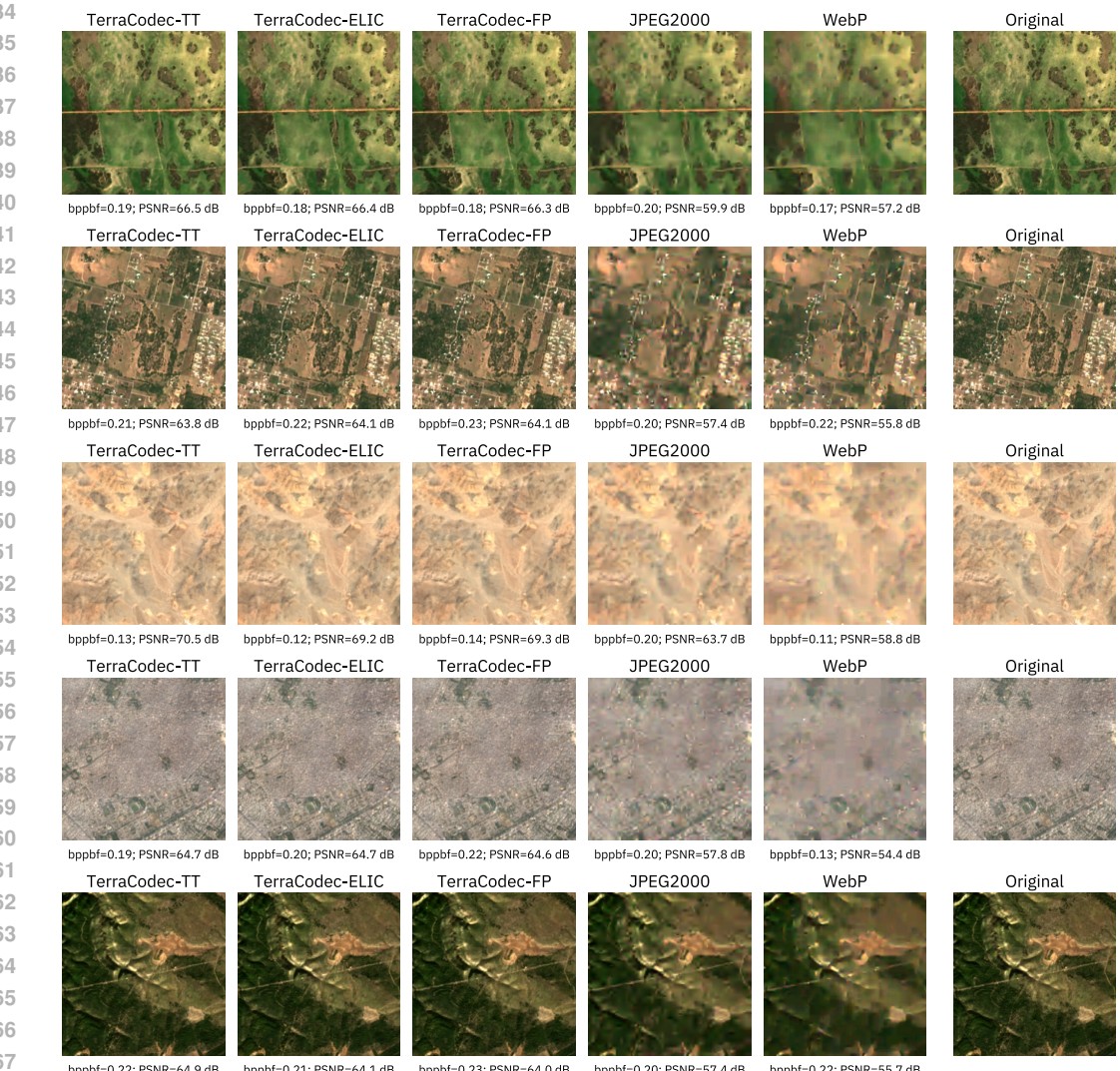

Figure 15: Reconstructions at $\approx$0.20 bppbf on SSL4EO-S12 v1.1.

**Experimental setup.** We follow the AllClear evaluation protocol, reporting metrics across all spectral bands (not per-band). We compare against heuristic baselines (LeastCloudy, Mosaicing) and zero-shot neural methods (CTGAN, DiffCR, PMAA, U-TILISE, UnCRtainTS). LeastCloudy selects the input image with the lowest cloud+shadow coverage, while Mosaicing fills each pixel by copying a single clear value, averaging if multiple are clear, or using 0.5 if none are clear. On AllClear, these heuristics rank among the top three zero-shot methods, outperforming most neural baselines without fine-tuning.

Besides reporting metrics on the full test set, we also evaluate subsets stratified by cloudiness. Difficulty thresholds are defined from the distribution of average cloud cover across the three input frames. Specifically, the 90th percentile (top 10%) corresponds to 0.99 average cloud cover, the 80th percentile (top 20%) to 0.78, and the 50th percentile (top 50%) to 0.49. For each sample, cloud cover is computed as the mean fraction of cloudy pixels across the three timestamps. Cloud masks are generated using the *s2cloudless* algorithm in Google Earth Engine and provided with the dataset.

We use TEC–TT's prior mean prediction ($\mu$) as a clear-sky estimate of the next frame, capturing large-scale structure while down-weighting transient noise such as clouds (App. F.2.2). Building on this, we adapt TEC–TT for cloud removal by predicting the third input frame from the two previous ones and applying *cloud-aware decoding*: clear regions retain their original tokens, while cloudy

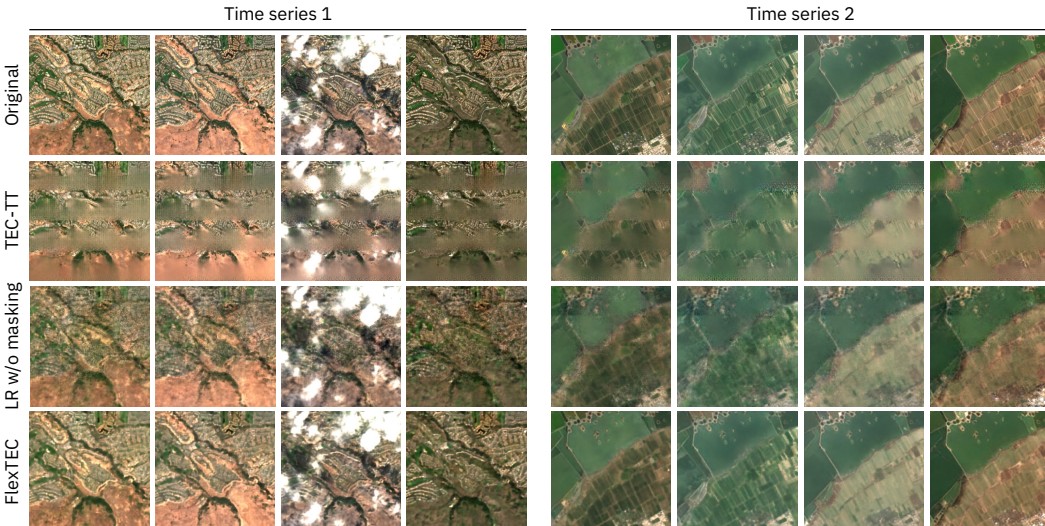

Figure 16: Effect of filling dropped tokens ($K{=}5$) with model prior predictions at inference for different TEC–TT variants. Vanilla TEC–TT exhibits spatial holes and banding when tokens are removed. Adding Latent Repacking without masking improves quality but leaves uneven detail, while FlexTEC preserves global layout and reduces artifacts.

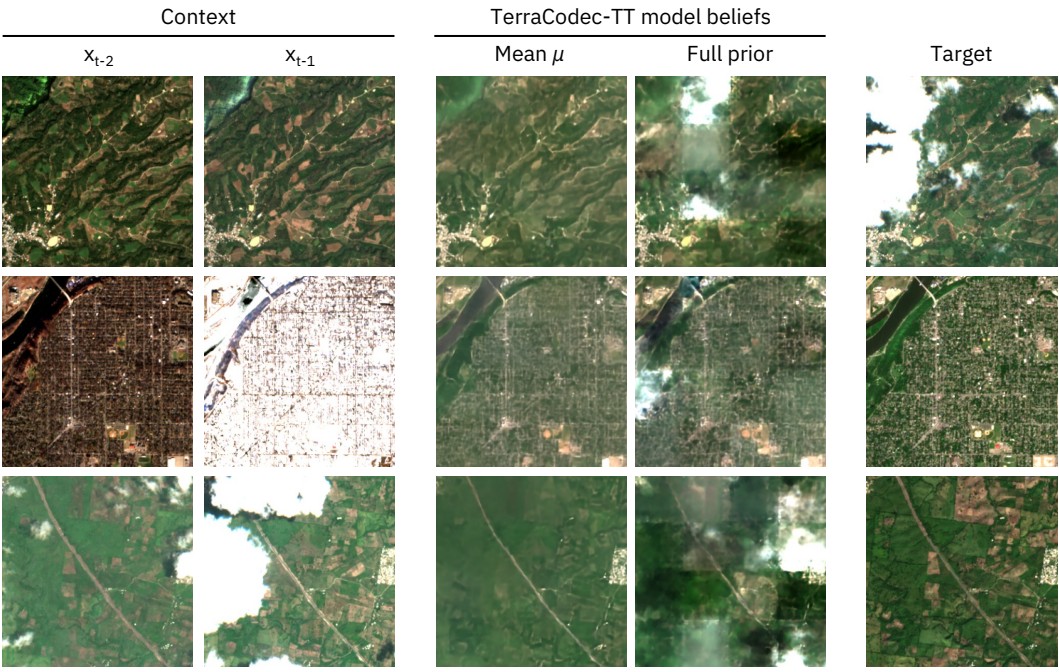

Figure 17: TEC–TT forecasts using only past context ($K = 0$). Model beliefs are visualized via the mean prediction ($\mu$) and full prior sampling from $\mathcal{N}(\mu, \sigma^2)$. The $\mu$-forecast reconstructs coherent large-scale structure, while sampling reveals plausible variations (e.g., clouds, surface texture) that reflect the model's uncertainty.

regions are replaced with the transformer's predicted tokens. Cloud and cloud-shadow masks are available and are downscaled to latent resolution using average pooling.

To provide the clean temporal context, we apply a *context reordering* heuristic that duplicates the least-cloudy inputs when few clean frames are available. We evaluate 16 variants spanning mask

type, mask threshold (0.0 vs. 0.5), context reordering (on vs. off), and decoding policy: *Interleave*, which predicts only cloudy tokens, and *Propagate*, which predicts from the first cloudy token onward, plus a *pure forecasting* baseline that replaces all tokens with predictions. The best setting uses cloud+shadow masks at threshold 0.0, average-pool downscaling, context reordering, and interleave decoding; all cloud-aware variants outperform pure forecasting, which tends to follow seasonal drift rather than reconstruct the target. Intuitively, conditioning only cloudy regions on predictions while preserving clear tokens reduces seasonal bias and allows TEC–TT to focus its temporal prior on reconstructing the true clear-sky target. Throughout the paper (Sec. 5.3, App. G.1), we report results using this best configuration.

Although our introduced codecs are trained on L2A reflectance data, the AllClear benchmark is defined on L1C inputs. To ensure a fair zero-shot evaluation, we therefore pretrained a TEC–TT model on the SSL4EO-S12 v1.1 L1C data modality using a high $\lambda = 700$ to focus on reconstruction quality. This model uses the same architecture, hyperparameters, and training procedure as the corresponding L2A variant, and is applied without any task-specific fine-tuning on AllClear.

## G.1 QUANTITATIVE RESULTS

Table 7: Test SSIM ($\uparrow$), RMSE ($\downarrow$), and MAE ($\downarrow$) on AllClear across difficulty subsets (by average cloudiness). Following the benchmark, the metrics are computed across all bands, not per-band.

| Model | 10% (hardest) | | | 20% | | | 50% | | | 100% (all) | | |
|---|---|---|---|---|---|---|---|---|---|---|---|---|
| | SSIM | RMSE | MAE | SSIM | RMSE | MAE | SSIM | RMSE | MAE | SSIM | RMSE | MAE |
| *Baseline heuristics* | | | | | | | | | | | | |
| LeastCloudy | 0.444 | 0.348 | 0.317 | 0.537 | 0.279 | 0.247 | 0.766 | 0.135 | 0.114 | 0.863 | 0.078 | 0.065 |
| Mosaicing | 0.107 | 0.162 | 0.136 | 0.183 | 0.162 | 0.131 | 0.558 | 0.101 | 0.075 | 0.755 | 0.062 | 0.045 |
| *Pre-trained models (zero-shot setting)* | | | | | | | | | | | | |
| CTGAN | 0.765 | 0.084 | 0.072 | 0.794 | 0.068 | 0.056 | 0.822 | 0.056 | 0.044 | 0.840 | 0.052 | 0.041 |
| DiffCR | 0.716 | 0.075 | 0.063 | 0.739 | 0.071 | 0.061 | 0.758 | 0.068 | 0.059 | 0.744 | 0.068 | 0.060 |
| PMAA | 0.746 | 0.071 | 0.060 | 0.758 | 0.076 | 0.066 | 0.770 | 0.080 | 0.071 | 0.768 | 0.086 | 0.078 |
| U-TILISE | 0.546 | 0.254 | 0.226 | 0.598 | 0.211 | 0.185 | 0.693 | 0.153 | 0.134 | 0.807 | 0.097 | 0.083 |
| UnCRtainTS | 0.813 | **0.061** | **0.046** | 0.826 | **0.063** | **0.049** | 0.865 | 0.057 | 0.044 | 0.898 | 0.050 | 0.039 |
| TerraCodec-TT | **0.814** | 0.064 | 0.050 | **0.830** | 0.065 | 0.050 | **0.887** | **0.045** | **0.034** | **0.917** | **0.034** | **0.025** |

Table 7 reports SSIM, RMSE, and MAE results for all methods, complementing the PSNR results in the main paper (Table 1). Our zero-shot TEC–TT clearly outperform the heuristics: TEC–TT reaches PSNR $\approx$ 32.9 dB and SSIM $\approx$ 0.917, compared to LeastCloudy (30.61 dB / 0.863) and Mosaicing (29.82 dB / 0.755). Relative to the strongest prior zero-shot neural method on All-Clear (UnCRtainTS, 29.01 dB / 0.898 / MAE = 0.039 / RMSE = 0.050), TEC–TT is substantially stronger (32.86 dB / 0.917 / MAE = 0.025 / RMSE = 0.034).

To provide further insight into the stratified evaluation, Figure 18 compares performance vs. cloudiness. While the heuristic baselines (Mosaicing, LeastCloudy) are competitive on less cloudy images, the figures clearly show how they struggle on the highly cloudy subsets, with performance notibally degradings. Zero-shot neural methods are more robust, though their performance also declines as past context becomes increasingly obscured.

## G.2 QUALITATIVE RESULTS

Figure 19 presents additional TEC–TT inpainting results on the AllClear test set across varying degrees of cloudiness in the three past input frames. While three context images are shown, TEC–TT uses only the two least cloudy as input. Notably, even under heavily clouded conditions, TEC–TT leverages past context to produce relatively accurate predictions of the target frame.

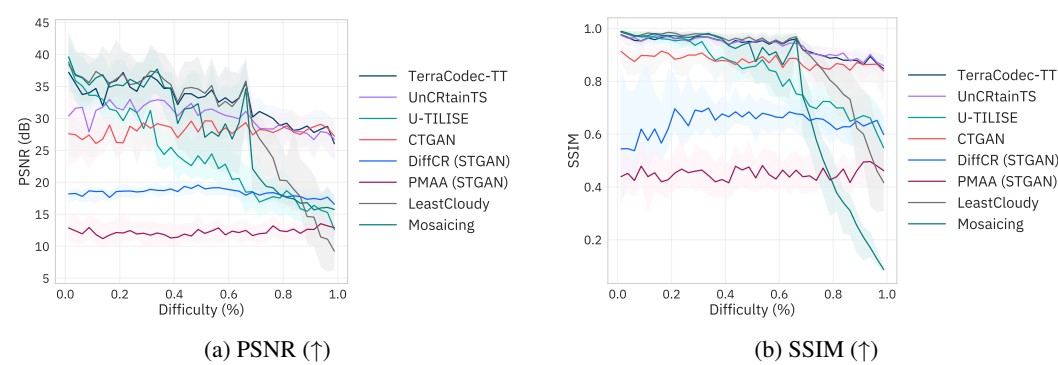

(a) PSNR (↑)          (b) SSIM (↑)

Figure 18: Performance vs. cloudiness on the AllClear test set. Each curve shows the median metric across samples binned by average cloud cover, with shaded ribbons indicating interquartile range (IQR).

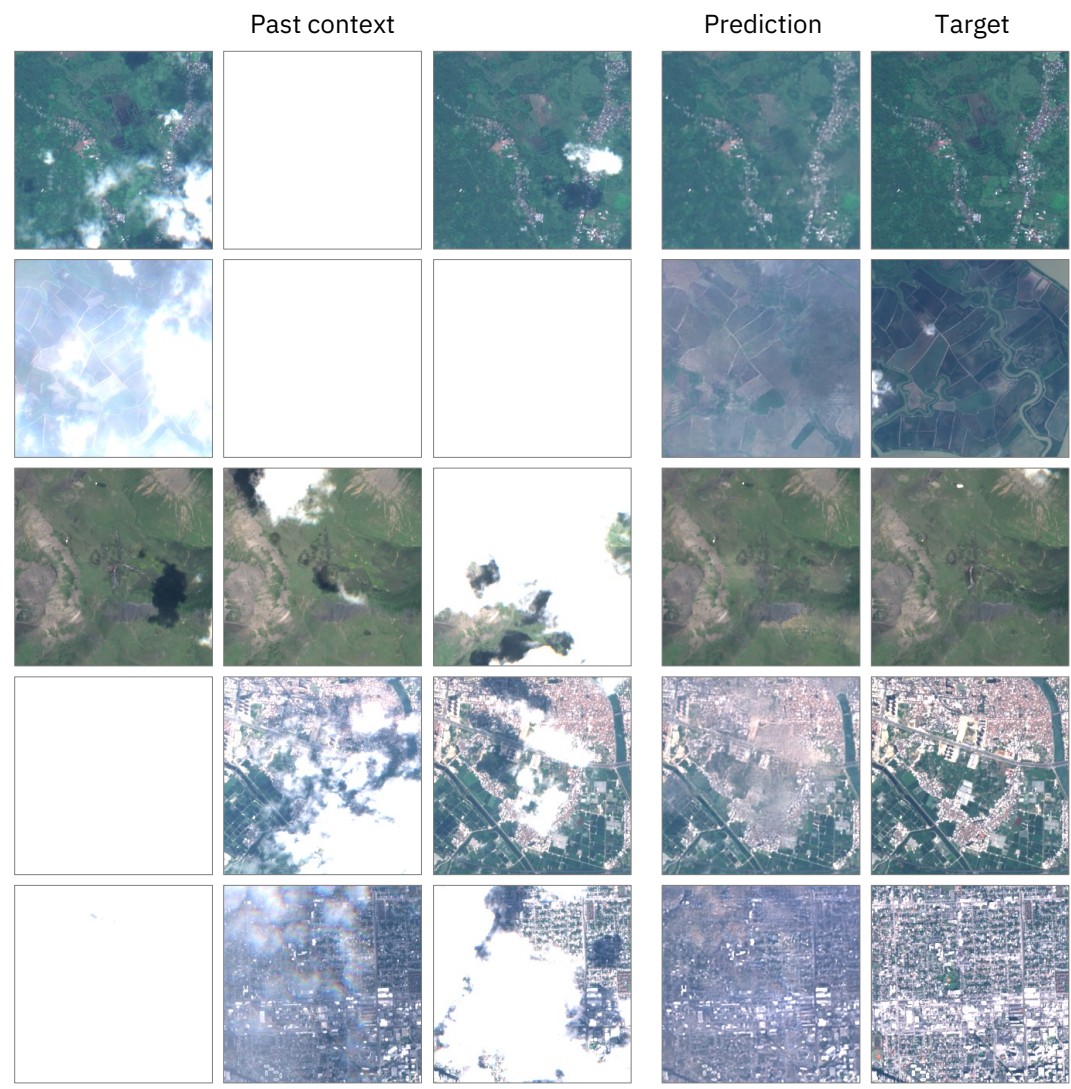

Figure 19: Cloud inpainting examples with TEC–TT on the AllClear benchmark. Reflectance values are clipped to 0–2000, which causes clouds to appear saturated in the visualizations.

