# OpenReview forum: "TerraCodec: Compressing Earth Observations"
_ICLR.cc/2026/Conference — Submitted to ICLR 2026_

### Official Review · Reviewer_XaXV · 2025-10-27

**Soundness:** 2
**Presentation:** 2
**Contribution:** 1
**Rating:** 2
**Confidence:** 5

**Summary:**

This paper propose TerraCodec, a series of neural codecs for EO data based on well-designed architectures. The author use Factorized Prior variant, ELIC, and VCT architecture and adapt them for multispectral and temporal EO images. What’s more, the author propose flexible-rate codec by selecting different latent channels for transmission. The proposed codecs outperform classical codes and can be utilized for downstream tasks and zero-shot cloud inpainting.

**Strengths:**

- The author consider the multispectral and temporal dependency of EO images.
- The writing and presentation of the paper are excellent, with a clear and logical flow.

**Weaknesses:**

- The proposed codecs are **lack of novelty**, with no specific design for EO images.
- More performance comparison tests of neural compression need to be included, such as [1,2,3]
- **The setting for downstream tasks evaluation is impractical.** To demonstrate that the compressed images have minimal impact on downstream tasks, previous works, including task-oriented compression, use well-trained AI models and tested with compressed images without finetuning. First, downstream tasks may involve images from various satellites with different codecs, making it impossible to fine-tune for a single codec's reconstructed images. Second, users may not have sufficient data for fine-tuning. Additionally, the authors don’t demonstrate that finetuning with TerraCodec-FP’s reconstructed images would not negatively affect the task accuracy for images from other sources.

[1]  Remote sensing image compression based on high-frequency and low-frequency components

[2] COSMIC: Compress Satellite Images Efficiently via Diffusion Compensation

[3] Map-Assisted Remote-Sensing Image Compression at Extremely Low Bitrates

**Questions:**

- Is there any specific design for EO images, considering multispectral, 16-bit inputs and temporal dependency, distinct from video? Or the proposed TerraCodec only change the input channels and pretrained on EO dataset?
- In line161, TEC-TT tries to model temporal dependencies of seasonal EO data, however, for LEO satellite, a single orbit revolution is only 90 minutes. Therefore, the dependency between EO data is not seasonal. The angle, jitter, and cloud cover factors of satellite shooting between time-series images should be considered more.
- In Figure6, why the results of TerraCodec-TT (image only) is not equal with TerraCodec-ELIC? When TEC-TT reduces to an image codec, the architecture is the same as ELIC.
- FlexTEC exhibits poor RD performance, even underperforming TerraCodec-FP at low bitrates. Previous work [4] has shown that better RD performance can be achieved when selecting different channels to achieve variable-rate.

[4] Slimmable Compressive Autoencoders for Practical Neural Image Compression

---

> ### Author Response · Authors · 2025-11-24
>
> We thank the reviewer for the constructive and detailed feedback, and for acknowledging the clear writing and consideration of multispectral and temporal dependencies. We have uploaded an updated version of the paper that incorporates clarifications and improvements (Highlighted in blue), and address each point below.
>
> > Novelty and EO-Specific adaptations: "The proposed codecs are lack of novelty."
>
> We clarify that we do not claim architectural novelty for the underlying codecs. Factorized Prior and ELIC are established architectures, and their use is intentional: the EO domain lacks large-scale pretrained multispectral codecs, with prior work often relying on small datasets and benchmarking against ImageNet-based RGB models. To address this, we release EO-pretrained multispectral variants of FP and ELIC to provide reproducible baselines.
>
> Beyond adjusting model dimensions, EO-specific adaptations include per-band standardization to handle highly non-uniform pixel distributions and computing the loss in standardized space to ensure balanced gradients across channels.
>
> Our scientific novelty lies in exploring temporal modelling for EO compression. TEC-TT is, to our knowledge, the first large-scale temporal learned codec for EO. The model builds on an established architecture chosen for its suitability for flexible latent-space modelling. Using established backbones allows us to isolate the key scientific question: the effect of temporal modelling for EO compression.
>
> > Comparisons to Neural Compression Methods ([1]–[3]): "More performance comparison tests of neural compression need to be included."
>
> Our work focuses on 12-band multispectral compression with large-scale pretraining (over 240k time-series samples, and up to 1M individual images for the image-based codecs). Comparing this directly to prior remote-sensing compression work is not possible due to the lack of released pretrained multispectral models, the RGB-only focus of many methods, and the small-scale evaluations used in existing multispectral work. Methods [2,3] operate on RGB data, [1] is trained and evaluated on very small multispectral datasets (below 5k samples, not S-2), has not been tried in a large-scale pretraining setting, and would encounter a distribution shift on Sentinel-2 data. This would be an unfair comparison, given that our models are pretrained on S-2 data.
>
> For this reason, we benchmark multispectral variants of two well established codecs (Factorized Prior and ELIC), trained under the same large-scale setup to clearly isolate the effects of temporal modelling for EO compression. We have updated our paper and the related work section to better clarify these gaps to existing work.
>
> > Downstream Evaluation Setup: "The setting for downstream tasks evaluation is impractical. To demonstrate that the compressed images have minimal impact on downstream tasks, previous works, including task-oriented compression, use well-trained AI models and tested with compressed images without finetuning. "
>
> Our downstream setup reflects how EO models are typically built: encoder–decoder pipelines are finetuned on task-specific datasets (With either frozen or unfrozen encoders) ([5-6]) and we follow this setup for both compressed-reconstructed and uncompressed data.
>
> Fine-tuning is not introduced to compensate for compression artefacts but is standard practice for training the downstream model. We acknowledge the assumption that either all task data is compressed-reconstructed or all uncompressed, and see this as consistent with current EO pipelines. Extending this analysis to mixed compressed/uncompressed datasets would likely introduce a distribution shift for the fine-tuned model, but we can include such an analysis if the reviewer considers it relevant. EO downstream models are typically sensor-specific, which matches our setup. We updated our paper to better clarify the downstream task analysis setup.
>
> [5] Marsocci et al., PANGAEA: A Global and Inclusive Benchmark for Geospatial Foundation Models
>
> [6] Lacoste et al., GEO-Bench: Toward Foundation Models for Earth Monitoring
>
> > Temporal dependency: seasonal vs short revisit intervals: "However, for LEO satellite, a single orbit revolution is only 90 minutes. Therefore, the dependency between EO data is not seasonal. "
>
> We use seasonal data to study a challenging long-gap setting where radiometry, land-cover state, and cloud cover vary significantly between frames. Shorter revisit intervals would increase temporal redundancy and likely benefit a TEC-TT pretraining on such data. Factors such as jitter and cloud cover are present in SSL4EO-S12, and the model learns to handle them directly in latent space. Appendix E.2 provides P-frame and missing-context ablations that quantify these temporal gains, and this analysis is also discussed in our responses to Reviewers 4KrB and ZAgu.

---

> ### Author Response · Authors · 2025-11-24
>
> > Relation between TEC-TT (image-only) and TEC-ELIC performance: "Why are the results of TerraCodec-TT (image only) not equal with TerraCodec-ELIC?"
>
> TEC-TT and TEC-ELIC share the same image encoder–decoder, but differ in latent modeling for entropy coding. ELIC uses a context+hyperprior, whereas TEC-TT uses a latent transformer. This difference explains the discrepancy in RD performance.
>
> > FlexTEC rate–distortion performance: "FlexTEC exhibits poor RD performance, even underperforming TerraCodec-FP at low bitrates. Previous work [4] has shown that better RD performance can be achieved when selecting different channels to achieve variable-rate."
>
> FlexTEC is based on a predictive latent mechanism: Variable-rate compression is achieved by dropping latent tokens, which the temporal model reconstructs during decoding. We treat TEC-TT with full latents as the upper bound and use FlexTEC to illustrate how strong latent models can enable flexible-rate compression by predicting missing latents. A direct comparison to the other TEC models is limited. They optimize a single RD setting while FlexTEC learns 16 settings within (nearly) the same compute budget, naturally leading to lower performance for a specific RD setting. Slimmable CAEs [4] vary the rate through channel-scaling in lightweight image codecs, which we agree is a relevant and orthogonal research direction. We refer to our answer “Latent Repacking” to reviewer 4KrB for a more detailed answer.

---

### Official Review · Reviewer_nojc · 2025-11-01

**Soundness:** 2
**Presentation:** 4
**Contribution:** 3
**Rating:** 4
**Confidence:** 5

**Summary:**

TerraCodec targets EO compression with three models (FP/ELIC/TT) and a single-checkpoint variable-rate variant (FlexTEC) via Latent Repacking. Experiments on Sentinel-2 show RD gains vs. JPEG/JPEG2000/WebP/HEVC, plus zero-shot cloud removal and downstream robustness.

**Strengths:**

The paper aligns well with practical EO data characteristics, focusing on multispectral and temporal redundancy and addressing real-world demands such as variable-rate compression and downstream validation. The overall engineering implementation is solid, with clear structure and detailed configurations. The topic is timely and relevant, and the work provides potential value for large-scale EO data storage and transmission scenarios.

**Weaknesses:**

The experimental design does not fully support the claimed scope. Validation is limited to Sentinel-2 and lacks evaluation on datasets with different spatial, spectral, and temporal resolutions, which raises questions about generalizability. Some experiments are not sufficiently convincing, particularly the absence of quantitative metrics such as end-to-end encoding and decoding latency. The choice of baselines is somewhat outdated, lacking comparisons with the latest codec standards and learned compression frameworks. In addition, the novelty appears relatively weak—TEC-FP and TEC-ELIC seem to be adaptations of existing architectures rather than newly proposed methods.

**Questions:**

1. Dataset and Scope Expansion: Please include additional EO datasets (e.g., fMoW, USMapping, Landsat, MODIS) to verify the method’s performance under different spatial, spectral, and temporal resolutions, demonstrating its generalization ability.

2. Baseline Completeness: Extend the comparisons to include VTM, JPEG XL, and other recent standards, as well as diffusion-based and INR-based learned compression methods. In addition, comparisons with state-of-the-art EO-specific compression models (e.g., HL-RSCompNet) are needed for fairness and completeness.

3. Algorithm Efficiency: Provide quantitative evaluations of end-to-end encoding and decoding efficiency, including runtime and resource usage. A speed–quality curve comparing TEC variants would help illustrate their practical advantages.

4. Downstream Task Validation: Beyond classification and segmentation, please include regression-based downstream tasks (e.g., NDVI or vegetation index estimation) to show the general usability of compressed data across different task types.

---

> ### Author Response · Authors · 2025-11-24
>
> We thank the reviewer for the constructive and detailed feedback, and for emphasizing the practical relevance of multispectral and temporal redundancy. We have uploaded an updated version of the paper that incorporates clarifications (Highlighted in blue), and address each point below.
>
> > Scope and generalizability: "Validation is limited to Sentinel-2 and lacks evaluation on datasets with different spatial, spectral, and temporal resolutions."
>
> We updated the title to "TerraCodec: Compressing Optical Earth Observation Data" and clarified the Sentinel-2 multispectral scope in the abstract.The architectures are sensor-agnostic and can be applied to other sensors. However, learned compression generally involves a trade-off between broad generality and adaptation to sensor-specific characteristics captured by the entropy model. In practice, we see value in sensor-specific, compression-efficient codecs, consistent with EO operational pipelines that are mostly sensor-specific. Extending TerraCodec to other sensors or datasets (Landsat, MODIS, fMoW, USMapping) is feasible through finetuning or retraining. We refer to our answer “Generality and scope / Application to other sensors” to reviewer ZAgu for additional detail.
>
> > Novelty clarification (TEC-FP / TEC-ELIC): "the novelty appears relatively weak—TEC-FP and TEC-ELIC seem to be adaptations of existing architectures."
>
> As summarized in our “Novelty and EO-Specific Design” answer to Reviewer XaXV, TEC-FP and TEC-ELIC intentionally use established backbones. We do not claim architectural novelty for these components. The gap we address is the lack of pretrained multispectral learned codecs for EO and the exploration of video-style compression for EO. We adapt FP and ELIC to 12-band multispectral data, pretrain on large-scale data and release all temporal and non-temporal models to support reproducible baselines and future research.
>
> > Baseline selection: "The choice of baselines is somewhat outdated."
>
> Our focus is multispectral (12-channel) compression. RGB-only codecs require re-engineering or retraining to operate on multispectral data. Most EO compression approaches do not release weights or code and are evaluated and trained on small datasets, limiting direct comparisons. As noted above, we therefore train multispectral variants of FP and ELIC as practical baselines. We agree that benchmarking INR-based and diffusion-based methods on large-scale, multispectral EO data is relevant future work.
>
> > Runtime and efficiency metrics: "absence of quantitative metrics such as end-to-end encoding and decoding latency."
>
> Thanks for pointing this out, we add encoding/decoding times, model sizes, and inference throughput for TEC-FP, TEC-ELIC, TEC-TT, and FlexTEC in an updated paper version. Please also see our response to Reviewer kPzn for a more detailed discussion of the intended offline deployment setting and potential transformer optimizations.
>
> > Downstream task validation: "Beyond classification and segmentation, please include regression-based downstream tasks."
>
> We agree that expanding downstream validation to regression-based tasks (e.g., NDVI or vegetation indices) is valuable. Our current downstream section already provides an initial look beyond RD metrics which is the main evaluation in compression literature. While we provide zero-shot cloud in-painting results and two different downstream tasks (classification and segmentation), our paper mainly focuses on the actual Sentinel-2 image compression. Classification and segmentation are the main downstream tasks in most EO benchmarks such as PANGAEA or GEO-bench.

---

### Official Review · Reviewer_4KrB · 2025-11-01

**Soundness:** 2
**Presentation:** 4
**Contribution:** 3
**Rating:** 4
**Confidence:** 4

**Summary:**

The paper presents TerraCodec, a family of learned codecs for multispectral and temporal Earth Observation (EO) imagery, including TEC-FP, TEC-ELIC, and the temporal Transformer-based TEC-TT. It further proposes FlexTEC, a single-checkpoint variable-rate scheme leveraging Latent Repacking. Experiments on Sentinel-2 demonstrate improvements over classical codecs (JPEG, JPEG2000, HEVC) and maintain downstream performance for tasks such as land-cover classification and flood mapping.

**Strengths:**

1. The work targets multispectral and temporal EO imagery, incorporating per-band normalization and temporal modeling, demonstrating good engineering practice for EO data.
2. The paper provides detailed model and training setups, improving reproducibility and practical applicability.
3. The proposed FlexTEC offers a practical mechanism to achieve variable bitrates within a single model, which is relevant for real EO applications.
4. The authors verify that compressed data remain useful for EO downstream tasks, demonstrating functional robustness.

**Weaknesses:**

1. The title and abstract suggest “Compressing Earth Observation” in general, but experiments are limited to Sentinel-2. The work would be more accurately described as “Sentinel-2 multispectral image compression.” Broader validation (e.g., Landsat, MODIS) would strengthen the generalization claim.
2. The related work section omits several influential recent studies, such as C3, PnVC (INR-based), and diffusion-based compression models. In addition, recent SOTA compression methods for remote sensing imagery are not discussed. Including both general and EO-specific works would make the review more complete.
3. Unclear contribution boundary:  TEC-FP and TEC-ELIC are adaptations of existing frameworks (Factorized Prior and ELIC) to EO imagery and should not be presented as novel contributions. The methodological innovation primarily lies in Latent Repacking/FlexTEC and in systematizing EO compression practice, not in the architectural variants themselves.
4. Incomplete experimental comparisons:  The study lacks modern baselines—neither the latest codec standard VVC (H.266) nor contemporary learned methods (e.g., diffusion-based or INR-based) are included. It also omits compression approaches in the remote sensing domain. Extending the comparisons to these would significantly improve credibility.

**Questions:**

1. The title and abstract suggest general EO compression, yet all experiments use Sentinel-2. Can the authors include results from at least one additional EO source (e.g., Landsat, MODIS) to support claims of generalization?
2. Are TEC-FP and TEC-ELIC introducing new algorithmic components or primarily applying existing compression backbones to EO data? Please delineate what is new beyond domain-specific adaptations.
3. The Related Work section omits influential recent studies such as C3, PNVC, and diffusion-based compression frameworks. Please update this section and re-situate TerraCodec in the current research landscape.
4.  The comparison set lacks modern codecs and neural approaches. Please include evaluations against VVC, C3, or diffusion-based compression methods, as well as recent SOTA compression methods for remote sensing imagery,  to ensure fair and up-to-date benchmarking.
5. Analysis No runtime or complexity metrics are provided. Please add encoding/decoding latency, throughput, and resource usage comparisons among TEC-EP/ELIC/TT/FLEX variants to assess deployment feasibility.
6. The “latent repacking” mechanism resembles existing flexible bitrate methods. Please specify the conceptual or technical differences and provide evidence of improved generality or efficiency.
7. Figure 1 lacks TEC-ELIC visual results. Please include side-by-side reconstructions.
8. It would be valuable to evaluate how different temporal reference settings affect reconstruction quality — including the number of reference frames, temporal intervals, degree of land-cover change, and cloud coverage ratio within the temporal context.

---

> ### Author Response · Authors · 2025-11-24
>
> We thank the reviewer for the constructive and detailed feedback, and for recognizing the alignment of our design with EO data characteristics. We have uploaded an updated version of the paper that incorporates clarifications, which are highlighted in blue, and address each point below.
>
> > Title and scope: "The title and abstract suggest general EO compression, yet all experiments use Sentinel-2"
>
> We updated the title to TerraCodec: Compressing Optical Earth Observation Data and clarified the Sentinel-2 multispectral scope in the abstract. As noted in our response to Reviewer ZAgu, the architectures and EO-specific adaptations are sensor-agnostic but require finetuning or retraining when transferring to new sensors. From a compression standpoint, sensor-specific codecs are valuable for operational EO pipelines, while multi-sensor compression remains relevant future work. We added a paragraph on this to the Limitations section.
>
> > Clarifying contribution boundaries (TEC-FP/ELIC): "Are TEC-FP and TEC-ELIC introducing new algorithmic components or primarily applying existing compression backbones to EO data?"
>
> Factoried Prior and ELIC are established architectures that we adapt and pretrain on large-scale multispectral EO data. We intentionally use existing backbones because the EO domain currently lacks openly available pretrained multispectral codecs (specifically for S-2). These models provide reproducible baselines and serve as reference points for us and the community to study the effect of temporal modelling in EO compression. We refer to our “Novelty and EO-Specific Design” response to Reviewer XaXV for additional detail.
>
> > Related work: "The Related Work section omits influential recent studies such as C3, PNVC, and diffusion-based compression frameworks."
>
> We extended the Related Work section and added references covering C3, PNVC/INR-based compression, diffusion-based compression, and additional EO-specific approaches. Thank you for pointing these out!
>
> > Experimental comparisons: "The comparison set lacks modern codecs and neural approaches."
>
> Our work focuses on multispectral, 12-channel compression from. Direct comparisons to RGB-pretrained models or methods designed for RGB inputs are not feasible for Sentinel-2 data and requires adaptations or retraining. Previous EO compression approaches are often RGB-only or do not release pre-trained weights/code, which limits comparability.
>
> We therefore trained multispectral variants of FP and ELIC, which serve as baselines for evaluating temporal compression. Benchmarking INR-based and other neural video compression approaches for multispectral EO is a valuable suggestion. The release of our models provides a foundation for such research.
>
> > Runtime and complexity metric: "No runtime or complexity metrics are provided."
>
> As outlined in our response to Reviewer kPzn, we will include encoding/decoding times, model sizes, and inference throughput for our models (TEC-FP, TEC-ELIC, TEC-TT, and FlexTEC).
>
> > Latent Repacking: "The “latent repacking” mechanism resembles existing flexible bitrate methods."
>
> Latent Repacking with masked training differs from existing flexible-rate approaches in that it leverages a learned entropy model for a latent inference mechanism: latents are partially dropped to flexibly reduce rate, and the latent model predicts the missing parts at decode time.
>
> To our knowledge, many flexible-rate methods adjust model size, decide how many latent channels to keep, apply spatial masking, or adapt quantization strength; they do not reconstruct/ infer missing latents during the decoding process. We view this approach as particularly suitable for architectures with strong latent models such as a temporal transformer.
>
> We do not claim broader generality or performance over existing methods, but see this as a relevant and complementary research direction for flexible-rate compression in settings with strong latent models. We added this clarification to the Limitations section.
>
> > Figure 1: "lacks TEC-ELIC visual results."
>
> We kept the first figure intentionally simple to highlight the comparison between image-based and temporal models, but we appreciate the feedback and added a side-by-side comparison of all models (including ELIC) in the updated version.
>
> > Temporal reference settings: "It would be valuable to evaluate how different temporal reference settings affect reconstruction quality."
>
> Temporal conditioning analyses, including a P-frame ablation and missing-context experiments, are provided and discussed in Appendix E.2. They show consistent gains from temporal modelling even under our challenging seasonal setting. Land-cover change and cloud-coverage analyses remain interesting future work; Appendix Figure 18 includes an experiment reporting cloud-inpainting reconstruction performance under varying degrees of cloudiness.

---

### Official Review · Reviewer_ZAgu · 2025-11-03

**Soundness:** 3
**Presentation:** 3
**Contribution:** 3
**Rating:** 6
**Confidence:** 4

**Summary:**

The paper presents TerraCodec (TEC), a family of neural compression models made specifically for Earth Observation (Sentinel-2 L2A) imagery. TEC is designed for multispectral time-series Earth observation data, where existing image codecs (like JPEG) fail to capture temporal redundancy, and video codecs rely on motion priors that are unsuitable for the radiometric evolution of largely static EO scenes.

TerraCodec includes:
- TEC-FP: a simple, efficient CNN-based codec using a factorized prior.
- TEC-ELIC: a stronger version using spatial–channel attention for better rate–distortion trade-offs.
- TEC-TT: a temporal transformer that learns relationships across time, capturing how EO scenes change seasonally rather than through motion.
- FlexTEC: a flexible-rate model based on TEC-TT, trained with Latent Repacking and masking, allowing users to control bitrate from a single checkpoint.

**Strengths:**

The paper offers a well-motivated and effective solution in Earth Observation data storage and transmission, offering a real-world impact accompanied by technical innovation. The work introduces temporal transformers and a flexible-rate compression mechanism through latent repacking. The experimental setup is comprehensive and carefully executed, covering rate–distortion performance, zero-shot cloud inpainting, and clear qualitative analyses. Moreover, the methodology is transparent and well-documented, with extensive implementation details and a commitment to open-source release, which enhances the work’s reproducibility and value to the community.

**Weaknesses:**

**Generality and scope**: It would be useful to evaluate the transferability of TerraCodec’s pretrained models to other sensors (e.g., Sentinel-1 SAR or Landsat-8) without retraining. If such a transfer is not feasible, clarifying the underlying technical limitations would strengthen the discussion. Moreover, assessing how TerraCodec performs on Sentinel-2 L1C imagery would be valuable, as it would enable direct application to datasets like SEN12MS-CR-TS [R1] for cloud inpainting.

**Computational cost**: Training TEC-TT and FlexTEC appears computationally demanding and may be impractical for non-research users. Including runtime or inference-time benchmarks would help assess the models’ operational viability.

**Limited temporal depth in cloud inpainting experiments**: Evaluation uses 4-frame sequences, shorter than real-world EO time series, so long-term compression benefits may be underrepresented.

**Downstream extensions**: The paper could further exploit the temporal modeling capacity of TEC-TT by exploring zero-shot change detection as an additional downstream task. Additionally, since many downstream EO tasks utilize only the RGB bands of the L2A product, exploring this setting could allow the model to interface with RGB-only datasets such as CloudTran++ [R2] for cloud-removal tasks.

### Overall assessment
This paper presents a well-motivated and technically solid contribution to learned compression for Earth observation data. The work is clearly presented and experimentally convincing. I recommend acceptance, although some aspects could be further investigated.

### Minor clarity issues:
- Fig. 3 could benefit from a clearer caption with a step-by-step flow explanations.
- A minor labeling inconsistency in Figure 9. “Context for TerraCodec-TT”: the second image is marked a x_(t-2) again, but it should likely be labeled as x_(t-1) to correctly represent the temporal sequence.

### References
[R1] P. Ebel et al., "SEN12MS-CR-TS: A Remote-Sensing Data Set for Multimodal Multitemporal Cloud Removal," in IEEE Transactions on Geoscience and Remote Sensing, vol. 60, pp. 1-14, 2022
[R2] Christopoulos et al., "CloudTran++: Improved Cloud Removal from Multi-Temporal Satellite Images Using Axial Transformer Networks". Remote Sensing, 17, 86, 2025

**Questions:**

- Can the method be applied to other sensors (e.g., Sentinel-1 SAR or Landsat-8) without retraining?
- Have the authors tried the method on Sentinel-2 L1C data? What is the expected performance?
- How efficient are the introduced models during inference?

---

> ### Author Response · Authors · 2025-11-24
>
> We thank the reviewer for the constructive and detailed feedback, for emphasizing the real-world impact, comprehensive experimental design, and value of open-sourcing TerraCodec. We have uploaded an updated version of the paper that incorporates clarifications (Highlighted in blue), and address each point below.
>
> > Generality and scope / Application to other sensors
>
> Our models are pretrained for Sentinel-2, but the architecture and training pipeline are sensor-agnostic. Applying TerraCodec to other sensors (e.g., Sentinel-1 SAR or Landsat-8) is possible but would require finetuning or retraining. A multisensor compressor would likely need more diverse training data, and the entropy model would be less tailored to a specific sensor’s statistics, which can affect compression performance. Given typical data volumes and EO operational pipelines, we see value in compressors fine-tuned to specific sensors. We clarified this and updated the title to TerraCodec: Compressing Optical Earth Observation Data.
>
> We also trained S-2 L1C variants (used in the cloud-inpainting experiment) and observe performance similar to L2A. We are open to releasing these models; the present release focuses on L2A.
>
> > Computational cost
>
> We will add encoding/decoding times, model sizes, and inference throughput for all models. As noted in our response to kPzn, TEC-TT and FlexTEC are best suited for offline storage and ground-segment transfer, while the lighter FP and ELIC variants also support edge use.
>
> > Limited temporal depth in cloud inpainting experiments
>
> Models trained with longer temporal context would likely reveal further temporal gains, but they also substantially increase computational cost, using two previous frames already provides a strong compression reduction and offers a practical balance between temporal context and efficiency. Our p-frame ablation (Figure 6) shows the temporal gains achievable with longer sequence evaluations within the current model. For cloud inpainting, we rely on an established benchmark dataset to ensure fair comparison.
>
> > Downstream extensions
>
> TEC-TT naturally lends itself to zero-shot change detection: learning the distribution of the “most likely next frame” allows deviations to be flagged directly as change or anomalies. Given the spatial structure of the latents, simple spatial change maps are also feasible. We see this as promising future work. For RGB-only tasks, RGB is obtained by selecting the corresponding bands from the reconstructed 12-band output. Our current models were not trained with RGB-only inputs, which would be required for RGB inpainting or cloud-removal tasks, but this would be a straightforward extension.
>
> > Minor clarity issues
>
> Thank you for pointing these things out. We updated Fig. 3’s caption to: “Each input image is first encoded into latents by an ELIC image encoder. The per-image latents are tokenized, and a temporal transformer models these tokens autoregressively, predicting the mean and scale parameters for the current frame’s tokens based on past latents.” We also fixed the labeling inconsistency in Fig. 9.

---

### Official Review · Reviewer_kPzn · 2025-11-03

**Soundness:** 3
**Presentation:** 3
**Contribution:** 3
**Rating:** 6
**Confidence:** 3

**Summary:**

This paper introduces TerraCodec, a family of learned compression models specifically designed for Earth observation (EO) data, with a focus on multispectral and multi-temporal satellite imagery. The core idea is to learn a compact latent representation of EO data using a transformer-based autoencoder and exploit temporal dynamics to improve compression efficiency and reconstruction fidelity. A key technical contribution is Latent Repacking, which enables flexible bitrate control without retraining. The method is evaluated on large-scale EO datasets and shows strong performance in both compression metrics (e.g., bits per pixel, BPP) and downstream tasks like cloud inpainting—specifically on the AllClear benchmark—without task-specific fine-tuning.

**Strengths:**

- The proposed Latent Repacking mechanism is elegant and practical. By dynamically truncating or quantizing latent codes based on importance (e.g., entropy or variance), the model supports continuous rate adaptation—a significant advantage over fixed-rate neural codecs.
- TerraCodec achieves state-of-the-art rate-distortion performance on EO data. More impressively, it demonstrates zero-shot generalization to downstream tasks

**Weaknesses:**

- While comparisons to image codecs and EO-specific baselines are provided, the paper does not benchmark against modern neural video codecs
- Transformer-based models can be computationally expensive. The paper lacks details on encoding/decoding latency, memory footprint, or inference speed—critical factors for deployment on ground stations or edge devices. A complexity vs. performance trade-off analysis would be valuable.

**Questions:**

- Can Latent Repacking be extended to hierarchical or spatially adaptive compression?

---

> ### Author Response · Authors · 2025-11-24
>
> We thank the reviewer for the constructive and detailed feedback, and for highlighting the elegance and practicality of Latent Repacking and the strong zero-shot generalization of TerraCodec. We have uploaded an updated version of the paper that incorporates clarifications (Highlighted in blue), and address each point below.
>
> > Benchmarking against neural video codecs: "While comparisons to image codecs and EO-specific baselines are provided, the paper does not benchmark against modern neural video codecs"
>
> A fair comparison to modern neural video codecs would require pretraining them on multispectral EO data (specifically Sentinel-2), which is computationally costly. As the current state of the art for EO compression is image-based, we pretrained and released learned image codecs (FP and ELIC) on multispectral EO for a fair benchmark against our temporal models. To our knowledge, our work is the first to train a video-style compression model on multispectral EO data. We agree that evaluating additional neural video codecs is relevant future work, and the release of our temporal models is intended to provide a foundation for this.
>
> > Computational Complexity and Operational Deployment: "A complexity vs. performance trade-off analysis would be valuable."
>
> We will add encoding/decoding times, model sizes, and inference throughput for all models (TEC-FP, TEC-ELIC, TEC-TT, and FlexTEC) in an updated paper version. We see the use case of larger transformer-based models like TEC-TT in offline, large-scale compression pipelines (e.g., data centers or ground-segment storage), where latency is less critical and RD performance is prioritized. The convolutional TEC-FP and TEC-ELIC variants are considerably lighter and more suitable for edge deployment. Furthermore, we also note that the underlying VCT architecture we build upon reports frame rates comparable to other modern neural video codecs (e.g., DCVC, FVC, ELF-VC).
>
> > Latent Repacking: “Can Latent Repacking be extended to hierarchical or spatially adaptive compression?”
>
> Latent Repacking with masked training is introduced for transformer-based latent spaces. Existing hierarchical or spatially adaptive codecs [1–4] typically rely on convolutional encoders with simpler latent models. While masking could in principle be applied there as well, we see it as particularly suited for models with strong latent predictors (such as a Temporal Transformer) where missing latents can be reliably reconstructed (see Fig. 9).
> For transformer-based latents, Latent Repacking can be extended to hierarchical setups by applying it separately to latent groups, or to spatially adaptive schemes by varying token/channel budgets across spatial regions. We see especially the latter as promising future work.
>
> [1] Minnen et al., Joint Autoregressive and Hierarchical Priors for Learned Image Compression, NeurIPS 2018.
>
> [2] Duan et al., Lossy Image Compression with Quantized Hierarchical VAEs, WACV 2023.
>
> [3] Song et al., Spatially-Adaptive Feature Transform for Variable-Rate Compression, ICCV 2021.
>
> [4] Tong et al., QVRF: Quantization-Error-Aware Variable-Rate Framework, ICIP 2023.

---

### Comment · Area_Chair_9Avj · 2025-11-25

Dear Reviewers,

Thank you for your time and effort in reviewing submissions for ICLR 2026. As we begin the author-reviewer discussion process, we kindly remind you to submit your responses to the author rebuttals by **December 2**.

Your engagement in this discussion phase is crucial to ensuring a fair and thorough evaluation of each submission.

### **Action Required**
- Carefully consider the authors’ rebuttal and any additional evidence they provide.
- Update your review (if applicable) to reflect your revised perspective.
- Discuss with the authors if further details are required

Your AC

---

### Meta-Review · Area_Chair_RAsa · 2026-01-15

**Summary:**

This paper has mixed initial reviews: 6, 6, 4, 4, 2. Many comments (comparison with more baseline methods, extension to other datasets, complexity analysis) are shared among the reviewers. However, no additional results are presented due to various difficulties. None of the reviewers responds to the authors' responses. Given that no additional results are presented, I would doubt that the current responses would be agreeable to all the reviewers.

Reviewer kPzn (6: marginally above the acceptance threshold; 3: You are fairly confident in your assessment.)

-	While comparisons to image codecs and EO-specific baselines are provided, the paper does not benchmark against modern neural video codecs

-	Transformer-based models can be computationally expensive. The paper lacks details on encoding/decoding latency, memory footprint, or inference speed—critical factors for deployment on ground stations or edge devices. A complexity vs. performance trade-off analysis would be valuable.

-	Can Latent Repacking be extended to hierarchical or spatially adaptive compression?

[AC: The first comment is included as future work. The second comment is NOT addressed properly. The authors indicate “We will add encoding/decoding times, model sizes, and inference throughput for all models (TEC-FP, TEC-ELIC, TEC-TT, and FlexTEC) in an updated paper version.” No results are provided at this point.]

Reviewer ZAgu (6: marginally above the acceptance threshold; 3: You are fairly confident in your assessment.)

-	Generality and scope: It would be useful to evaluate the transferability of TerraCodec’s pretrained models to other sensors (e.g., Sentinel-1 SAR or Landsat-8) without retraining. If such a transfer is not feasible, clarifying the underlying technical limitations would strengthen the discussion. Moreover, assessing how TerraCodec performs on Sentinel-2 L1C imagery would be valuable, as it would enable direct application to datasets like SEN12MS-CR-TS [R1] for cloud inpainting.

-	Computational cost: Training TEC-TT and FlexTEC appears computationally demanding and may be impractical for non-research users. Including runtime or inference-time benchmarks would help assess the models’ operational viability.

[AC: Same concern as kPzn, but no results are presented.]

-	Limited temporal depth in cloud inpainting experiments: Evaluation uses 4-frame sequences, shorter than real-world EO time series, so long-term compression benefits may be underrepresented.

[AC: The authors argue that 2 frames offers a better balance between performance and complexity. No additional results are presented for 4 frames.]

-	Downstream extensions: The paper could further exploit the temporal modeling capacity of TEC-TT by exploring zero-shot change detection as an additional downstream task. Additionally, since many downstream EO tasks utilize only the RGB bands of the L2A product, exploring this setting could allow the model to interface with RGB-only datasets such as CloudTran++ [R2] for cloud-removal tasks.

[AC: This comment is more or less future work suggestions.]

Reviewer 4KrB (4: marginally below the acceptance threshold; 3: You are fairly confident in your assessment.)

-	The title and abstract suggest “Compressing Earth Observation” in general, but experiments are limited to Sentinel-2. The work would be more accurately described as “Sentinel-2 multispectral image compression.” Broader validation (e.g., Landsat, MODIS) would strengthen the generalization claim.

[AC: The comment is editorial and not so critical.]

-	The related work section omits several influential recent studies, such as C3, PnVC (INR-based), and diffusion-based compression models. In addition, recent SOTA compression methods for remote sensing imagery are not discussed. Including both general and EO-specific works would make the review more complete.

[AC: The comment is editorial and not so critical.]

-	Unclear contribution boundary:  TEC-FP and TEC-ELIC are adaptations of existing frameworks (Factorized Prior and ELIC) to EO imagery and should not be presented as novel contributions. The methodological innovation primarily lies in Latent Repacking/FlexTEC and in systematizing EO compression practice, not in the architectural variants themselves.

-	Incomplete experimental comparisons:  The study lacks modern baselines—neither the latest codec standard VVC (H.266) nor contemporary learned methods (e.g., diffusion-based or INR-based) are included. It also omits compression approaches in the remote sensing domain. Extending the comparisons to these would significantly improve credibility.

[AC: The authors argue that it may not be easy to do such comparisons because “Previous EO compression approaches are often RGB-only or do not release pre-trained weights/code.” At this point, no additional results are presented.]

-	Analysis No runtime or complexity metrics are provided. Please add encoding/decoding latency, throughput, and resource usage comparisons among TEC-EP/ELIC/TT/FLEX variants to assess deployment feasibility.

[AC: Same concern as two other reviewers, but no results are presented.]

-	It would be valuable to evaluate how different temporal reference settings affect reconstruction quality — including the number of reference frames, temporal intervals, degree of land-cover change, and cloud coverage ratio within the temporal context.

[AC: Same concern as another reviewer, but no results are presented.]
Reviewer nojc (4: marginally below the acceptance threshold; 5: You are absolutely certain about your assessment.)

-	Dataset and Scope Expansion: Please include additional EO datasets (e.g., fMoW, USMapping, Landsat, MODIS) to verify the method’s performance under different spatial, spectral, and temporal resolutions, demonstrating its generalization ability.

[AC: The authors articulate difficulties (limited data volumes, the need of re-training/fine-tuning) in testing their method on additional EO datasets.]

-	Baseline Completeness: Extend the comparisons to include VTM, JPEG XL, and other recent standards, as well as diffusion-based and INR-based learned compression methods. In addition, comparisons with state-of-the-art EO-specific compression models (e.g., HL-RSCompNet) are needed for fairness and completeness.

[AC: Same concerns as the other reviewers. No additional results are presented.]

-	Algorithm Efficiency: Provide quantitative evaluations of end-to-end encoding and decoding efficiency, including runtime and resource usage. A speed–quality curve comparing TEC variants would help illustrate their practical advantages.
[AC: Same concerns as the other reviewers. No additional results are presented.]

-	Downstream Task Validation: Beyond classification and segmentation, please include regression-based downstream tasks (e.g., NDVI or vegetation index estimation) to show the general usability of compressed data across different task types.

[AC: Limited additional results are provided. This comment does not seem to have been addressed.]

Reviewer XaXV (2: reject, not good enough; 5: You are absolutely certain about your assessment.)

-	The proposed codecs are lack of novelty, with no specific design for EO images.

[AC: The response looks okay to me.]

-	More performance comparison tests of neural compression need to be included, such as [1,2,3]

[AC: The authors argue that comparison with these prior works may not be fair due to the differences in input modality (RGB) or data volume. No additional results are presented.]

-	The setting for downstream tasks evaluation is impractical. To demonstrate that the compressed images have minimal impact on downstream tasks, previous works, including task-oriented compression, use well-trained AI models and tested with compressed images without finetuning. First, downstream tasks may involve images from various satellites with different codecs, making it impossible to fine-tune for a single codec's reconstructed images. Second, users may not have sufficient data for fine-tuning. Additionally, the authors don’t demonstrate that finetuning with TerraCodec-FP’s reconstructed images would not negatively affect the task accuracy for images from other sources.

[AC: No additional results presented. The current response is somewhat handwaving.]

**Reviewer Concerns:**

See the summary section.

**Reviewer Scores:**

This paper has mixed initial reviews: 6, 6, 4, 4, 2. Many comments (comparison with more baseline methods, extension to other datasets, complexity analysis) are shared among the reviewers. However, no additional results are presented due to various difficulties. None of the reviewers responds to the authors' responses. Given that no additional results are presented, I would doubt that the current responses would be agreeable to all the reviewers.

---

### Decision · Program_Chairs · 2026-01-26

Reject